# Epidermal resident memory T cell fitness requires antigen encounter in the skin

Eric S Weiss[1,2], Toshiro Hirai[3,4], Haiyue Li[1,2], Andrew Liu[1,2], Shannon Baker[1,2], Ian Magill[5,6], Jacob Gillis[1,2], Youran R Zhang[1,2], Torben Ramcke[1,2], Kazuo Kurihara[1,2], The ImmGen Consortium OpenSource T cell Project, David Masopust[7], Niroshana Anandasabapathy[8], Harinder Singh[2,9], David Zemmour[5,6], Laura K Mackay[10], Daniel H Kaplan[1,2]*

[1]Department of Dermatology, University of Pittsburgh, Pittsburgh, United States; [2]Department of Immunology, University of Pittsburgh, Pittsburgh, United States; [3]Institute for Microbial Diseases, Osaka University, Osaka, Japan; [4]Institute for Open and Transdisciplinary Research Initiatives, Osaka University, Osaka, Japan; [5]Department of Immunology, Harvard Medical School, Boston, United States; [6]Department of Pathology, Brigham and Women's Hospital, Harvard Medical School, Boston, United States; [7]Center for Immunology, Department of Microbiology and Immunology, University of Minnesota, Minneapolis, United States; [8]Department of Dermatology, Meyer Cancer Center, Program in Immunology and Microbial Pathogenesis, Weill Cornell Medicine, New York, United States; [9]Center for Systems Immunology, University of Pittsburgh, Pittsburgh, United States; [10]Department of Microbiology and Immunology, The University of Melbourne, The Peter Doherty Institute for Infection and Immunity, Melbourne, Australia

*For correspondence: dankaplan@pitt.edu

## eLife Assessment

This manuscript advances the prior finding that antigen recognition in the skill helps establish skin resident memory in CD8 T cells by elucidating the role of TGFBRIII in regulating CD8+ TRM skin persistence upon topical antigen exposure. Key novelty of the your work lies in generation and use of the CD8+ T cell-specific TGFBRIII knockout model, which allows them to demonstrate the role of TGFBRIII in fine tuning the degree of CD8+ T cell skin persistence and that TGFBRIII expression is promoted by CD8+ TRM encountering their cognate antigen upon initial skin entry. This is an **important** finding and is supported by **convincing** evidence. There are concerns about the use of FTY720 and the need to establish active TGFbeta limiting conditions to further test this working model.

**Abstract** CD8$^+$ tissue-resident memory T cells (T$_{RM}$) develop from effectors that seed peripheral tissues where they persist providing defense against subsequent challenges. T$_{RM}$ persistence requires autocrine TGFβ transactivated by integrins expressed on keratinocytes. T$_{RM}$ precursors that encounter antigen in the epidermis during development outcompete bystander T$_{RM}$ for TGFβ resulting in enhanced persistence. ScRNA-seq analysis of epidermal T$_{RM}$ revealed that local antigen experience in the skin resulted in an enhanced differentiation signature in comparison with bystanders. Upon recall, T$_{RM}$ displayed greater proliferation dictated by affinity of antigen experienced during epidermal development. Finally, local antigen experienced T$_{RM}$ differentially expressed TGFβRIII, which increases avidity of the TGFβRI/II receptor complex for TGFβ. Selective ablation of *Tgfbr3* reduced local antigen experienced T$_{RM}$ capacity to persist, rendering them phenotypically like bystander T$_{RM}$. Thus, antigen-driven TCR signaling in the epidermis during T$_{RM}$ differentiation results

in a lower TGFβ requirement for persistence and increased proliferative capacity that together enhance epidermal T_{RM} fitness.

## Introduction

Tissue-resident CD8 memory T cells (T_{RM}) are a highly abundant, non-circulating, long-lived subset of memory T cells that play an important role in protecting against re-infections (*Jiang et al., 2012*; *Peng et al., 2021*). T_{RM} also provide immunosurveillance against neoplasia and are thought to be pathogenic in some autoimmune diseases (*Oguejiofor et al., 2015*; *Virassamy et al., 2023*; *Strobl et al., 2020*; *Strong Rodrigues et al., 2018*; *van den Boorn et al., 2009*; *Zhang et al., 2023*; *Schunkert et al., 2021*). In the skin, following infection with vaccinia virus (VV) or herpes simplex virus, T_{RM} develop from CD8$^+$ T cell effectors (T_{EFF}) that expand following priming in the lymph node and are recruited into the skin by inflammatory signals where they preferentially reside in the epidermis (*Jiang et al., 2012*; *Allan et al., 2003*; *Bedoui et al., 2009*; *Liu et al., 2006*; *Reynoso et al., 2019*; *Gebhardt et al., 2011*; *Hirai et al., 2020*). Unlike T_{RM} in some tissues, re-encounter with cognate antigen in the skin is not required for T_{RM} differentiation (*McMaster et al., 2018*; *Lee et al., 2011*; *Mackay et al., 2012*; *Masopust et al., 2004*). Epidermal T_{RM} that encounter cognate antigen in the skin or bystander T_{RM} that do not encounter antigen both develop with comparable efficiency and both express similar levels of the canonical T_{RM} markers CD103 and CD69 (*Park et al., 2018*; *Hirai et al., 2021*).

The cytokine transforming growth factor β (TGFβ) is required for cutaneous T_{RM} development at multiple stages. TGFβ signaling in the lymph node during steady state epigenetically preconditions naïve CD8 T cells allowing for later T_{RM} differentiation (*Mani et al., 2019*). T_{EFF} recruited into skin require TGFβ signaling for entry into the epidermis and differentiation into T_{RM} (*Mackay et al., 2013*; *Mackay et al., 2015*). Once epidermal T_{RM} have differentiated, these cells continue to require TGFβ signaling to retain epidermal residence. TGFβ is produced bound to the latency-associated peptide that prevents bioactivity until the complex is activated, which in the epidermis is mediated exclusively by activation via the integrins α_vβ_6 and α_vβ_8 expressed by keratinocytes (*Aluwihare et al., 2009*; *Yang et al., 2007*; *Worthington et al., 2011*). Epidermal persistence of T_{RM} depends on autocrine TGFβ, which is transactivated by the integrins αvβ6 and αvβ8 (*Hirai et al., 2021*; *Hirai et al., 2019*). Inducible ablation in T_{RM} of a required component of the TGFβ receptor (TGFβRII) or TGFβ1, as well as ablation of αvβ6 and αvβ8 in keratinocytes, all result in loss of epidermal T_{RM} (*Hirai et al., 2021*; *Mohammed et al., 2016*).

We previously reported that local antigen experienced T_{RM} that have encountered cognate antigen in the skin are better able to persist in the epidermis than bystander T_{RM} that have not encountered antigen in the skin when active TGFβ is limited, despite the fact that both groups of cells had prior activation within the lymph node (*Hirai et al., 2021*). This is evident when TGFβ activation is experimentally reduced by small molecule inhibition of αvβ6 and αvβ8 and when established T_{RM} compete with newly recruited T_{EFF} cells for limiting amounts of TGFβ activation. In both instances, bystander T_{RM} are preferentially lost from the epidermis, while local antigen experienced T_{RM} persist. Thus, an encounter with antigen in the skin results in more fit T_{RM} and represents a potential opportunity to preferentially enrich antigen-specific over bystander T_{RM} cells during repeated challenges.

Herein, we delineate mechanisms of T_{RM} homeostasis by demonstrating that within a population of endogenous T_{RM}, antigen encounter in the skin is required for their full differentiation, whereas bystander T_{RM} are maintained at an earlier developmental stage. We also find that local antigen experienced T_{RM} have increased proliferative capacity during a recall response compared to bystander T_{RM} that is dependent on affinity of antigen experienced in skin, thus providing an additional functional attribute defining T_{RM} fitness. Finally, we show that expression of TGFβRIII by local antigen experienced T_{RM} which can be induced following TCR ligation is required for epidermal persistence when active TGFβ is limiting.

## Results

### Epidermal T_{RM} are transcriptionally heterogeneous

The recruitment of effector CD8$^+$ T cells into mouse flank skin via a viral infection (e.g. Vaccinia virus) or through an inflammatory stimulus (e.g. 'DNFB-pull') results in comparable numbers of long-lived

**eLife digest** We are constantly exposed to a wide array of pathogens in our environment. This would be detrimental to our survival if it were not for the immune system's ability to adapt. Each time the body encounters pathogens, it develops an immunological memory that allows it to mount a quick defense response. Recent research has shown that immunological memory is not confined to immune organs like lymph nodes and spleen. Instead, a large proportion is maintained in peripheral tissues at sites of prior infection.

These local cells, known as tissue-resident memory T cells ($T_{RM}$), provide the first line of defense against repeated infections. $T_{RM}$ develop from circulating precursors of memory T cells after pathogen exposure and then permanently reside in these tissues. The ability to mount pathogen-specific responses is a hallmark of immunological memory. Therefore, knowing how prior antigen exposure shapes $T_{RM}$ development is critical for understanding peripheral immunity. However, the signals that drive T cells to become $T_{RM}$ remain incompletely understood.

Using an acute viral infection model in mice, Weiss et al. investigated how local infection affects the differentiation and function of $T_{RM}$ in the skin. $T_{RM}$ cells were generated through skin infection with a poxvirus strain, and immune responses were measured using immunofluorescence, flow cytometry, and approaches that distinguish circulating from resident T cells.

The results showed that T cells must encounter pathogen-derived antigens directly in the skin, in addition to in the lymph nodes, for the effective development of $T_{RM}$, which are capable of mounting stronger recall responses. Their long-term survival in the skin depended on signaling through the transforming growth factor-β (TGFβ) pathway, which is activated by skin cells and enhanced in T cells that encountered pathogens within the epidermis.

$T_{RM}$ play important roles in cancer surveillance, pathogen clearance and vaccine response, but they can also contribute to disease when dysregulated, as seen in conditions such as psoriasis, vitiligo and graft-versus-host disease. Understanding the factors that promote TRM fitness may enable strategies for making these cells more effective in fighting infections. Further, TGFß represents a possible therapeutic target to selectively modulate $T_{RM}$ activity when these cells become harmful.

---

$CD103^+$ epidermal $T_{RM}$ and is independent of the presence or absence of cognate antigen in the skin (*Mackay et al., 2012*; *Hirai et al., 2019*; *Davies et al., 2017*). This, however, has only been demonstrated using HSV-specific (gBT-I) or OVA-specific (OT-I) TCR transgenic T cells. To determine if endogenous polyclonal $CD8^+$ T cells share this phenotype, we employed a dual VV infection 'DNFB-pull' model. Cohorts of wild-type C57BL/6 mice were infected on the left flank with VV by skin scarification (*Figure 1A*). The infection expanded $CD8^+$ effectors specific for VV antigens, which were then recruited into the left flank in response to the ongoing infection-induced inflammation. DNFB is applied to the right flank 5 days post-infection to recruit into the skin circulating CD8 effectors expanded by infection. Mice were then rested for 50+ days to allow for the formation of $T_{RM}$. We have previously found that evaluation of $T_{RM}$ numbers in the epidermis by flow cytometry is much less accurate than direct immunofluorescent visualization (*Hirai et al., 2021*). Evaluation of whole-mounted epidermal sheets stained with anti-CD8 revealed comparable numbers of $CD8^+$ T cells at the VV-infected (left flank) and DNFB-treated (right flank) sites (*Figure 1B and C*). Expression of CD103 on $T_{RM}$ as evaluated by flow cytometry was also similar at both sites (*Figure 1D* and *Figure 1—figure supplement 1A*). To evaluate the frequency of antigen-specific T cells at each site, skin from a separate cohort was examined by flow cytometry using the B8R tetramer which recognizes an immunodominant epitope of VV in $H2-K^b$ (*Moutaftsi et al., 2006*). We observed equivalent numbers of $B8R^+$ $T_{RM}$ at the VV-infected and DNFB-treated sites (*Figure 1E and F*). From these data, we conclude that polyclonal CD8 T cells form epidermal $T_{RM}$ with comparable efficiency when recruited into the skin by viral infection or sterile inflammation and that the presence or absence of cognate antigen in the skin has no effect on the number or frequency of antigen-specific $T_{RM}$.

Although $T_{RM}$ efficiently populate the epidermis independently of cutaneous cognate antigen, we have previously demonstrated that $T_{RM}$ that form at skin sites containing cognate antigen are functionally distinct from bystander $T_{RM}$ that form in its absence (*Hirai et al., 2021*). To analyze potential transcriptional differences between local antigen experienced and bystander $T_{RM}$, we performed

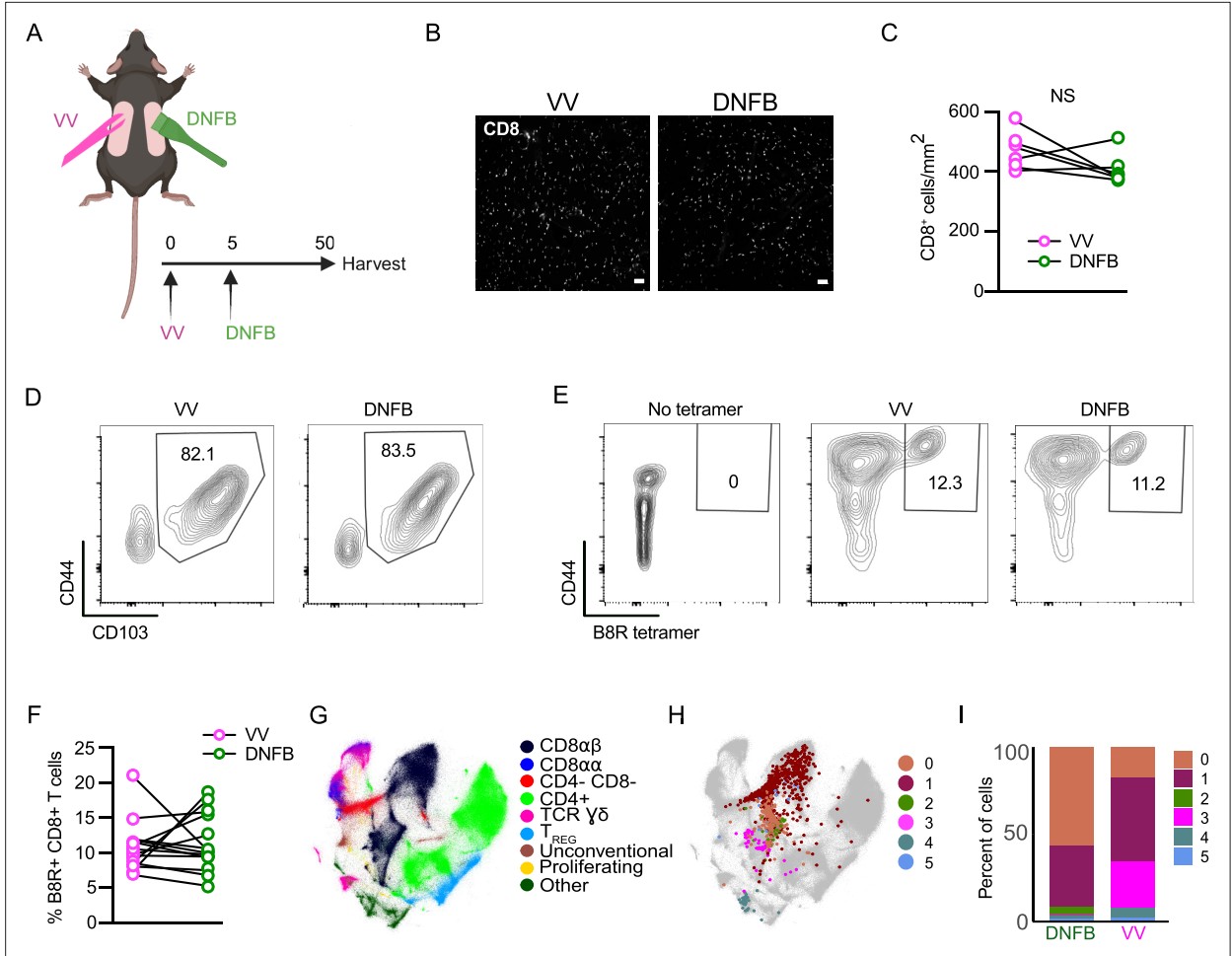

**Figure 1.** Epidermal T$_{RM}$ are transcriptionally heterogeneous. (**A**) Experimental design. Mice were treated with vaccinia virus infection by skin scarification on the left flank on day 0, and then on day 5 post-infection, the right flank was painted with 0.15% DNFB. On day 56, flanks were harvested for either epidermal whole mount, flow cytometry, or cells were sorted for scRNA-seq. (**B**) Representative images and (**C**) quantification of epidermal whole mounts of VV and DNFB treated flanks harvested on day 50 post infection stained for CD8a. (**D**) Representative flow plots gated on CD45$^+$ CD3$^+$ CD8$^+$ CD90.2$^+$ cells isolated from VV or DNFB treated flanks. (**E**) Representative flow plots and (**F**) quantification of B8R tetramer binding of CD103+ T$_{RM}$ gated as in (**D**) isolated from VV or DNFB treated flanks. (**G**) The integrated Minimal-Distorted Embedding of all 96 experiments from the ImmgenT consortium with annotated clusters. (**H**) Minimal-Distorted embedding visualization of transcriptional clusters of skin T$_{RM}$ projected over the ImmgenT dataset. (**I**) The percentage of cells in each transcriptional cluster found in VV or DNFB treated sites. Each symbol represents paired data from the same individual animal (**C, F**). Data shown to be nonsignificant by paired Student's t-tests (**C, F**). Data are representative of 3 separate experiments. Scale bar in (**B**) represents 50 µm. Panel A was created with BioRender.com.

The online version of this article includes the following figure supplement(s) for figure 1:

**Figure supplement 1.** Flow cytometric and gene expression analysis of the T$_{RM}$ used in the Immgen T scRNA-seq dataset.

single-cell RNA-seq in collaboration with the ImmgenT consortium (***Zemmour et al., 2022***). The consortium consists of multiple groups that isolated populations of murine T cells from multiple tissues and contexts and then subjected them to single-cell RNA-seq (***Zemmour et al., 2022***)(see Methods). 627,692 mature T cells from 80 different experiments and 703 samples were integrated and batch-corrected using totalVI (***Korsunsky et al., 2019***) (see Methods). Cells were projected in two dimensions using Minimal-Distorted Embedding (see Methods), revealing the expected transcriptional clustering of distinct types of T cells (***Figure 1G***). As part of the consortium, we isolated cells from the flanks of rested (>65 days) VV-infected or DNFB-treated flank skin by enzymatic digestion. Cells were pooled from 10 mice and purified based on expression of CD90 and CD8 (***Figure 1—figure supplement 1B***). As expected, most cells expressed CD69 and CD103 (***Figure 1—figure supplement 1C***). Skin cells from each group and naïve spleen cells were hash tagged and analyzed by single-cell

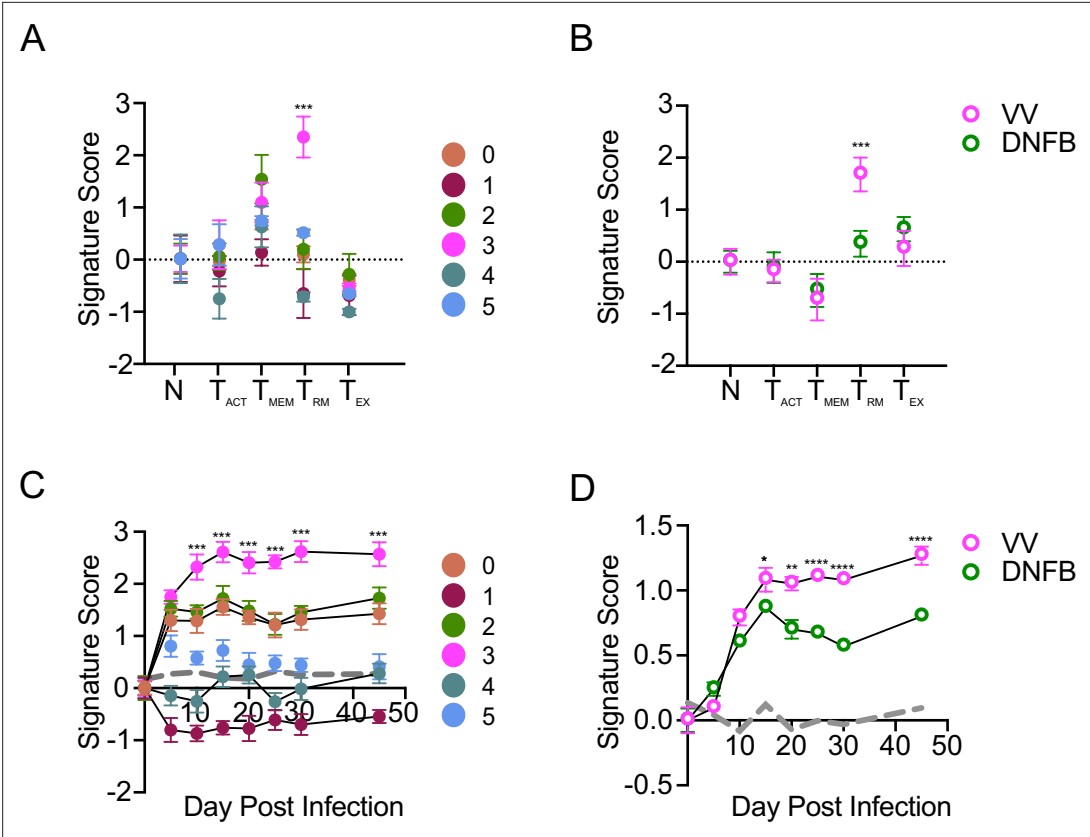

**Figure 2.** Cutaneous antigen is required for complete T<sub>RM</sub> differentiation. (**A**) Signature score analysis of skin T cell clusters calculating the enrichment for previously published T cell state gene sets (N=naïve). Signature score = Mean (average upgenes z-scores) - Mean(average downgenes z-scores). *** $P<0.001$ by Dunnett's multiple comparisons of cluster 3 to all other clusters. (**B**) Pseudobulk analysis comparing signature scores of cells isolated from VV and DNFB sites as in (**A**). ** $P<0.01$ and *** $P<0.001$ by paired Student's t-tests. (**C**) Signature score analysis of individual clusters or (**D**) cells isolated from VV and DNFB sites calculating the enrichment for compared to epidermal T<sub>RM</sub> isolated at the indicated day post infection. ***$P<0.001$ by Dunnett's multiple comparison test of cluster 3 to all other clusters in (**C**). * $P<0.05$, ** $P<0.01$, **** $P<0.0001$ by paired Student's t-tests in (**D**) Gray line represents a randomized control generated by the average enrichment of each group compared to a randomly generated gene set of an equal number of probes. Datapoints and error bars represent mean and 95% confidence interval of the relative enrichment of each set of DEGs.

RNA-seq. Analysis of transcripts revealed a total of 6 clusters (0–5) which are shown overlayed on the complete ImmgenT cell embedding (*Figure 1H*, *Figure 1—figure supplement 1D*). As expected, most cells clustered with CD8αβ T cells (*Figure 1G*, black). There was, however, a high degree of heterogeneity, and some possibly contaminating cells fell outside of this region. A comparison of the relative frequency of cells in each of the clusters (0–5) isolated from the VV or DNFB sites revealed a strong enrichment of cluster 3 in the VV group and of clusters 0 and 2 in the DNFB group (*Figure 1I*, magenta). This suggested that cluster 3 cells were manifesting a distinct transcriptional state likely induced by antigen encounter in the skin.

## Cutaneous antigen is required for complete T<sub>RM</sub> differentiation

To test the hypothesis that cells in cluster 3 could be T<sub>RM</sub> that have encountered antigen in the skin during development, we performed signature score enrichment analysis comparing the transcriptional profile of our cells to published core genes of well-defined CD8 T cell states. Signature scores were calculated based on normalized differential gene expression compared to core genes of T<sub>RM</sub> generated by acute infection models (GSE47045), activated T cells (T<sub>ACT</sub>, GSE10239), circulating memory T cells (T<sub>MEM</sub>, GSE41867), and exhausted T cells (T<sub>EX</sub>, GS41867) (*Jaiswal et al., 2022*; *Sarkar et al., 2008*; *Doering et al., 2012*). Notably, cells in cluster 3 showed enriched expression of T<sub>RM</sub> core genes compared to all other clusters (*Figure 2A*). We performed a similar analysis using pseudo-bulk analysis of all cells isolated from the VV or DNFB sites. Consistent with the enrichment of cluster 3 in cells from

the VV site, we found a strong enrichment of T$_{RM}$ core genes in the VV group (*Figure 2B*). These data suggest that cells in cluster 3 isolated from the VV site represent fully differentiated T$_{RM}$.

To further test the hypothesis that cluster 3 represents fully differentiated T$_{RM}$, we performed signature score enrichment analysis of clusters 0–5 against a dataset of skin T cells isolated at different times following skin VV infection (GSE79805) (*Pan et al., 2017*). Cells from clusters 0, 2 and 3 showed similar signature scores in comparison with skin T cells up to day 5 post-infection; however, clusters 0 and 2 then plateaued suggesting a lack of continued differentiation (*Figure 2C*). In contrast, cluster 3 transcripts scored higher at later time points in T$_{RM}$ differentiation. Signature scores for cells isolated from the DNFB and VV sites were similar at early time points but diverged at later time points with increased scores observed in cells from the VV site (*Figure 2D*). Taken together, these data suggest that T$_{RM}$ isolated from skin sites where they can encounter cognate antigen are transcriptionally more fully differentiated T$_{RM}$ while bystander T$_{RM}$ isolated from a site lacking cognate antigen remain in a less differentiated state.

## Local antigen experienced T$_{RM}$ have improved expansion in a recall response

Analysis of the 10 highest differentially expressed genes (DEGs) in cluster 3 (*Figure 1—figure supplement 1D*) showed increased expression of genes associated with T cell activation, including *Dusp1* and *Nr4a1* as well as the AP-1 family members J*unb* and *Fos* (*Papavassiliou and Musti, 2020*; *Owens and Keyse, 2007*; *Gazon et al., 2017*; *Schnoegl et al., 2023*; *Sun et al., 2021*). Given the importance of AP-1 family members in T cell proliferation and differentiation, we next tested whether this corresponded to functional differences in T$_{RM}$ activity (*Schnoegl et al., 2023*; *Atsaves et al., 2019*; *Shaulian and Karin, 2002*; *Buquicchio et al., 2024*). An important functional property of T$_{RM}$ is their capacity to rapidly expand following re-encounter with cognate antigen (*Szabo et al., 2019*; *Schenkel and Masopust, 2014*). To determine whether local antigen experienced and bystander T$_{RM}$ have different recall responses, we repeated our dual VV infection and DNFB 'pull' model in combination with an OT-I adoptive transfer model to allow for antigen recall using the SIINFEKL peptide. CD90.1$^+$ OT-I cells were adoptively transferred into naïve C57BL/6 mice followed by infection with a vaccinia virus expressing SIINFEKL peptide, OVA$_{257-264}$ (VV-OVA) on the left flank. On day 5 post-infection, the right flank was treated with DNFB to recruit expanded OT-I effectors to the site (*Figure 3A*). On day 50+ after infection, topical SIINFEKL peptide or control PBS was painted on both flanks in a single application (1° recall). The total number of OT-I cells in the epidermis 6 days later was determined by immunofluorescent microscopic evaluation of the CD90.1 congenic marker in epidermal whole mounts (*Figure 3B–C*). Restimulation with peptide led to an expansion of T$_{RM}$ at both sites compared to PBS-treated controls. Notably, local antigen experienced OT-I T$_{RM}$ cells in the VV-OVA treated flanks expanded to a larger extent compared to bystander T$_{RM}$ in DNFB treated flanks. To inhibit potential recruitment of circulating effector OT-I cells, we repeated these experiments administering FTY720 to block migration of T cells from lymph node or titrated anti-Thy1.1 antibody to ablate circulating OT-I but not T$_{RM}$ (*Figure 3D*). Both methods successfully depleted OT-I from the blood (*Figure 3—figure supplement 1A–D*), without affecting T$_{RM}$ in the skin (*Figure 3—figure supplement 1E*). Administration of FTY720 (*Figure 3E–F*) or anti-Thy1.1 treatment (*Figure 3G*) had no effect on the number of OT-I T$_{RM}$ in the epidermis after a 1° recall response and local antigen experienced T$_{RM}$ from the VV-OVA treated flank still expanded to a greater degree than bystander T$_{RM}$ from the DNFB treated flank. Thus, the increased number of epidermal OT-I at the local antigen experienced VV site during a 1° recall does not appear to result from recruitment of circulating cells.

Finally, to determine whether increased expansion by local antigen experienced T$_{RM}$ persists with multiple rounds of stimulation, we repeated this experiment but allowed mice to rest for 120 days after the 1° recall response. The number of T$_{RM}$ at both the VV-OVA and DNFB sites contracted down to equivalent numbers that were increased compared to PBS-treated, control mice (*Figure 3C*). Mice were then given a 2° recall by a second application of peptide at day +176. The number of T$_{RM}$ expanded but T$_{RM}$ at the VV-OVA site increased by a larger amount than T$_{RM}$ at the DNFB site (*Figure 3B–C*). Based on these data, we conclude that T$_{RM}$ encountering antigen in the skin during development expand more efficiently in response to antigen re-encounter and this phenotype persists long-term with subsequent antigen encounters.

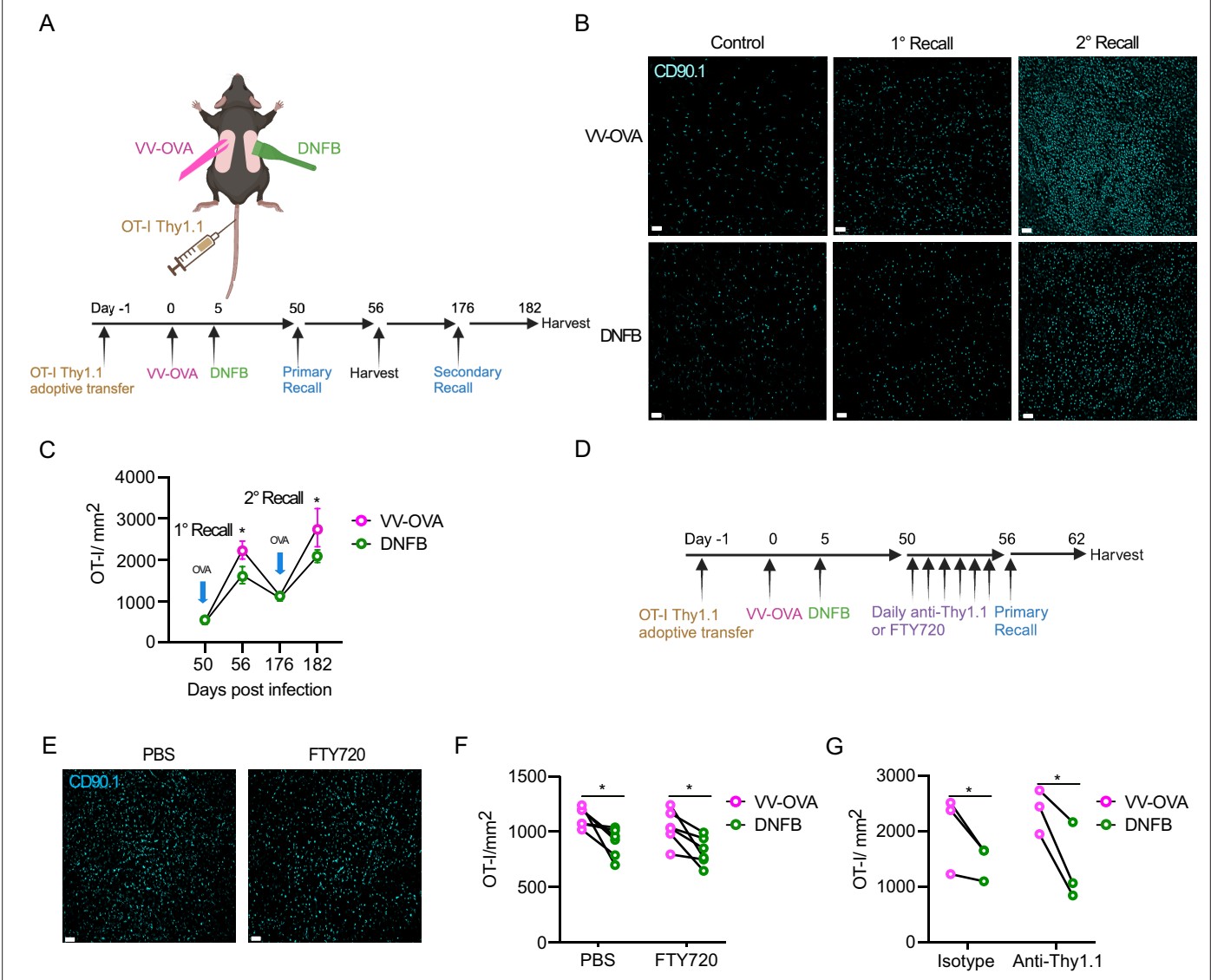

**Figure 3.** Local antigen experienced T$_{RM}$ have improved expansion in a recall response. (**A**) Experimental design. Mice were adoptively transferred with Thy1.1$^+$ OT-I T cells on day –1, then infected with OVA-expressing vaccinia virus by scarification on the left flank on day 0. On day 5 post-infection, the right flank was painted with 0.15% DNFB. On day 50, a primary recall response with SIINFEKL peptide in acetone and olive oil was painted on both flanks and harvested on day 56. In some cohorts, mice were allowed to rest for an additional 120 days and then treated with topical SIINFEKL peptide again on day 176 and harvested on day 182. (**B**) Representative images and (**C**) quantification of epidermal whole mounts isolated from VV-OVA and DNFB treated flanks stained for anti-Thy1.1 (n=10 animals). (**D**) Experimental design. Mice were treated as in (**A**), but for 6 days prior to SIINFEKL treatment, they were given either i.p. FTY720, i.p. PBS, i.v. titrated anti-Thy1.1 or i.v. isotype control. (**E**) Representative epidermal whole mount images and (**F**) quantification of skin from VV-OVA treated flanks on day 6 of a primary recall response treated with either FTY720 or PBS. (**G**) Quantification of total Thy1.1 OT-I cells in epidermal whole mounts on day 6 after a primary recall response in mice treated with either isotype or anti-Thy1.1 depleting antibody. Each symbol represents paired data from an individual same animal (**F, G**). Data are representative of 3 independent experiments. *$P<0.05$ by paired Student's t-tests. Unpaired Student's t-tests between FTY720 treated (**F**) or Anti-Thy1.1 treated (**G**) flanks and PBS treated flanks shows non-significance for both VV-OVA and DNFB. Scale bar represents 50 µm (**B, F**). Each symbol represents the mean +/- SEM (**C**). Panel A and D were created with BioRender.com.

The online version of this article includes the following figure supplement(s) for figure 3:

**Figure supplement 1.** FTY720 and anti-Thy1.1 deplete circulating OT-I cells withouth affecting the epidermis.

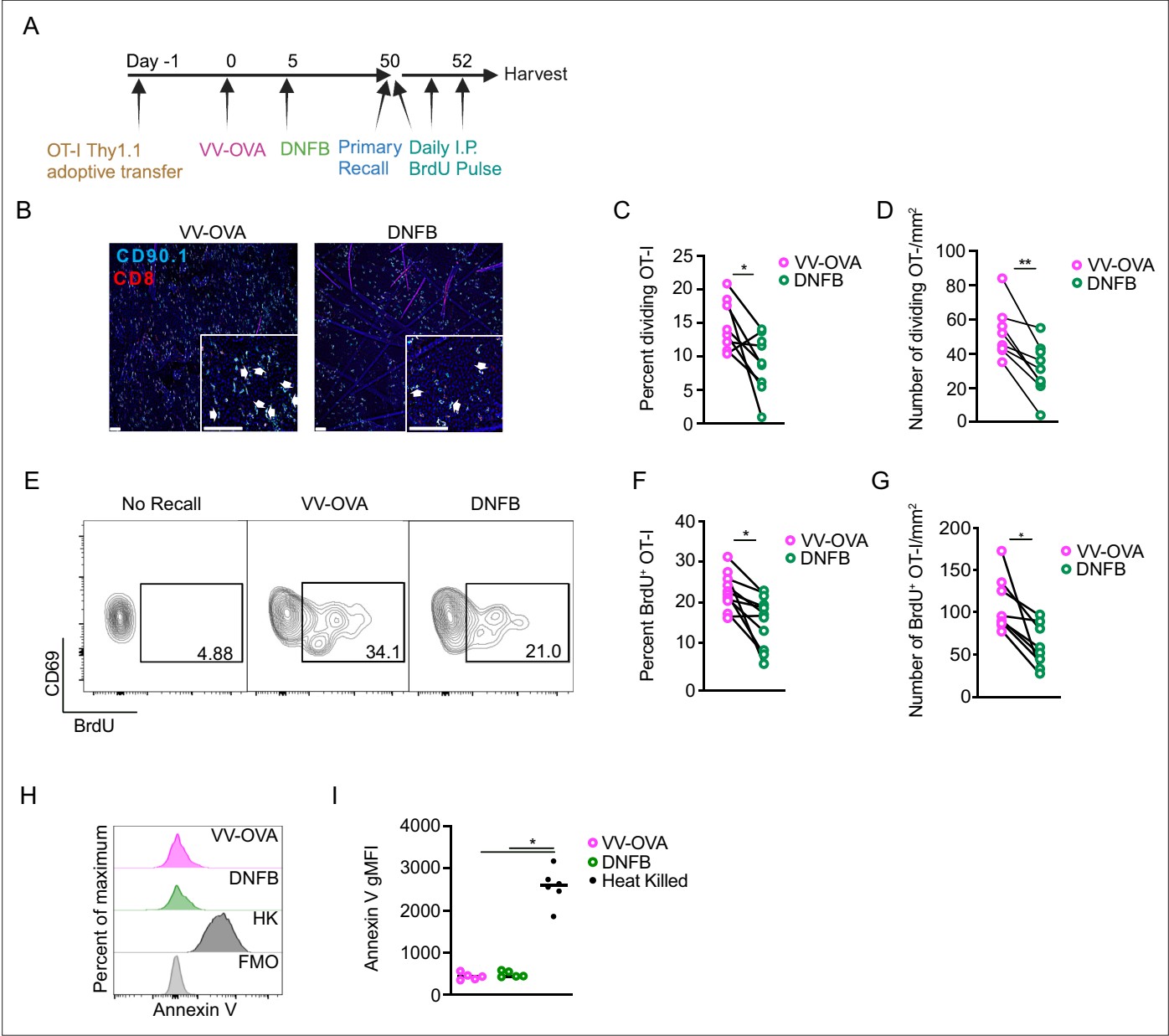

**Figure 4.** Local antigen experienced T$_{RM}$ have increased proliferation during a recall response. (**A**) Experimental scheme. (**B**) Representative images of epidermal whole mounts of VV-OVA or DNFB treated flanks on day 2 of a recall response. Arrows highlight cell doublets. (**C**) Quantification showing percent OT-I cells that are dividing in epidermal whole mounts on day 2 after primary recall response, and (**D**) total number of dividing OT-I cells. (**E**) Representative flow cytometric plots and (**F**) quantification showing BrdU incorporation in gated CD45$^+$ CD8$^+$ CD90.1$^+$ CD103$^+$ CD69$^+$ cells isolates from VV-OVA or DNFB treated flanks. (**G**) Quantification of total numbers BrdU$^+$ OT-I cells combining OT-I numbers with percentage of BrdU incorporation in (**F**). (**H**) Representative histograms and (**I**) quantification of Annexin V expression in OT-I cells isolated from VV-OVA or DNFB treated flanks or OT-I cells heat-treated at 60°C for 1 hr (HK). Data are representative of three separate experiments. *$P<0.05$, **$P<0.01$ by Student's paired t-tests (**C**, **D**, **F** and **G**) or Dunnett's test (**I**). Scale bar represents 50 µm. Panel A was created with BioRender.com.

## Local antigen experienced T$_{RM}$ exhibit increased proliferation during a recall response

We next hypothesized that the increased expansion of local antigen experienced T$_{RM}$ during a recall response resulted from increased proliferation. To test this, as above, WT mice adoptively transferred with OT-I cells were infected with VV-OVA on the left flank and DNFB on the right flank 5 days later. After at least 50 days of rest, both sides were painted with topical SIINFEKL peptide and mice were administered 2 mg of BrdU i.p. daily (**Figure 4A**). Epidermal sheets for immunofluorescent visualization

were harvested 2 days later. We noted a clear increase in the percent and total number of dividing cells as well as the total number of OT-I T cells in the epidermis at the VV-OVA site compared with the DNFB site (*Figure 4B–D*). Similar results were obtained by flow cytometry, with an increase in the percentage of BrdU$^+$ in T$_{RM}$ isolated from the VV-OVA site (*Figure 4E–G*). Staining for Annexin V was equivalent in both groups, suggesting that an altered rate of apoptosis does not contribute to the observed changes in expansion (*Figure 4H–I*). In sum, local antigen experienced T$_{RM}$ show augmented proliferation during an antigen recall response compared with bystander T$_{RM}$. Moreover, an enhanced proliferative capacity following antigen restimulation can now be added to other parameters of T$_{RM}$ fitness, including enhanced epidermal persistence when active TGFβ is limited.

## T$_{RM}$ fitness depends on TCR signal strength

We next hypothesized that the strength of TCR signal provided in the skin during T$_{RM}$ differentiation should correlate with T$_{RM}$ phenotype at late time points. To generate T$_{RM}$ with a range of local antigen encounter signal strengths, we wanted to expose newly-recruited T cells to either topical SIINFEKL peptide (N4), or a topical altered peptide ligand (APL) SIYNFEKL (Y3) that has ¼ the avidity for the TCR receptor (*Zehn et al., 2009*). We hypothesized that encountering an APL would induce an intermediate functional phenotype between full-strength SIINFEKL and no local antigen encounter. To test this hypothesis, WT mice were adoptively transferred with OT-I cells, then infected with VV-OVA on the left flank and 5 days later were challenged with DNFB on the right flank (*Figure 5A*). One day later, the DNFB site was painted with PBS, SII**N**FEKL (N4), or an APL SI**Y**NFEKL (Y3). After 50+ days of rest, all mice were challenged in a 1° recall response with topical N4 or control PBS at the DNFB sides (*Figure 5A*). Epidermal sheets for immunofluorescence were harvested 6 days later. As expected, mice restimulated with PBS showed equivalent numbers of T$_{RM}$ regardless of which peptide was provided during development (*Figure 5B–C*). Following a 1° recall response, T$_{RM}$ exposed to N4 (DNFB +N4) expanded to a level equivalent to T$_{RM}$ from the VV-OVA site. In contrast, T$_{RM}$ exposed to Y3 (DNFB +Y3) showed a degree of expansion that was intermediate compared to T$_{RM}$ that did not experience antigen in the skin (DNFB +PBS) and those that experienced SIINFEKL (DNFB +N4). Thus, the strength of TCR engagement determines the fitness of T$_{RM}$ based on their capacity to expand during a 1° recall response.

## Local antigen experienced T$_{RM}$ have improved persistence mediated by TGFßRIII

A feature of local antigen experienced T$_{RM}$ is that they are better able to persist in the epidermis than bystanders when levels of activated TGFβ are limited, either artificially or during competition with newly recruited T$_{EFF}$ (*Hirai et al., 2021*). We have previously found that expression of the canonical TGFβ receptors, *Tgfbr1* and *Tgfbr2* have equivalent expression in local antigen experienced and bystander T$_{RM}$ (*Hirai et al., 2021*). However, there is a third TGFβ receptor, TGFβRIII, that lacks a signaling component but functions as a reservoir of ligand for TGFβ, increasing avidity of the TGFβ receptor (*Miyazono, 1997*; *Heldin and Moustakas, 2016*). Notably, expression of TGFβRIII has been reported to increase in T cells following TCR ligation (*Ortega-Francisco et al., 2017*). This was confirmed in vitro with anti-CD3 anti-CD28 stimulated OT-I splenocytes (*Figure 6A*). We also observed increased expression of TGFβRIII in vivo, comparing local antigen experienced OT-I T$_{RM}$ with bystanders (*Figure 6B–C*). Based on this data, we hypothesized that increased expression of TGFβRIII on antigen experienced T$_{RM}$ could explain their capacity to maintain epidermal residence when available TGFβ is limiting.

To test this, we generated OT-I Thy1.1$^+$ E8i-cre$^{ERT2}$ huNGFR *Tgfbr3*$^{fl/fl}$ (Tgfbr3$^{ΔCD8}$) mice which allow for inducible ablation of *Tgfbr3* from T$_{RM}$ as well as control OT-I Thy1.1$^+$ E8i-cre$^{ERT2}$ huNGFR *Tgfbr3*$^{WT/WT}$ (Tgfbr3$^{WT}$) mice (*Figure 6D*). To validate these mice, Tgfbr3$^{ΔCD8}$ mice were treated with 0.05 mg/g tamoxifen i.p. for 5 days. CD8$^+$ T cells from tamoxifen-treated Tgfbr3$^{ΔCD8}$ mice were then isolated from spleen and lymph nodes and cultured for 2 days with anti-CD3 anti-CD28. Cells were evaluated by flow cytometry for expression of TGFβRIII and huNGFR, an indicator of successful induction of cre-mediated excision by tamoxifen. In the huNGFR-negative population where cre was not activated, approximately 50% of the cells expressed TGFβRIII (*Figure 6E–F*). In contrast, TGFβRIII was largely absent from cells co-expressing huNGFR, indicative of efficient ablation of *Tgfbr3*.

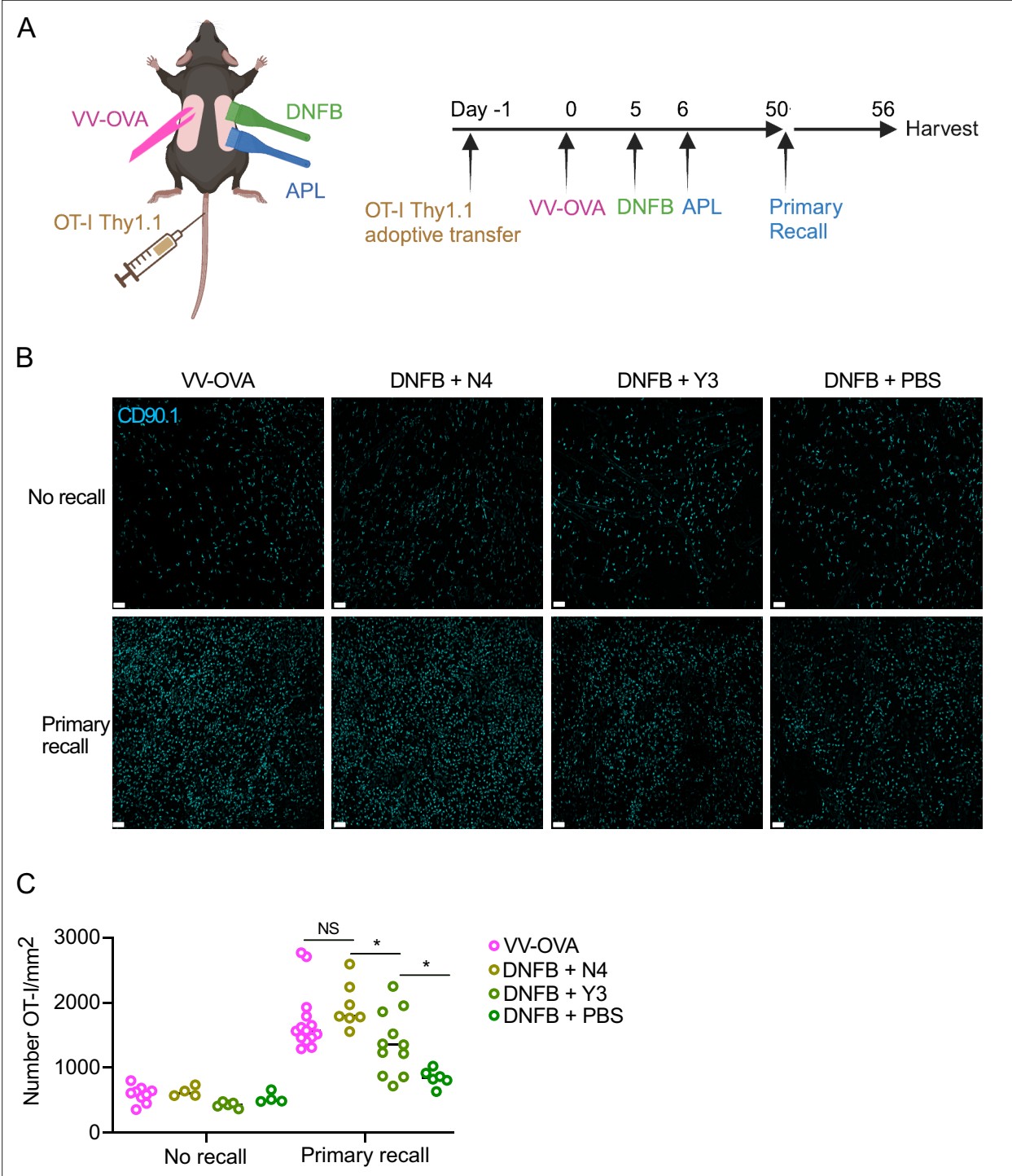

**Figure 5.** $T_{RM}$ fitness depends on TCR signal strength. (**A**) Experimental scheme. Local antigen experienced and bystander $T_{RM}$ were generated and restimulated as previously described, but on the DNFB-treated flanks, altered peptide ligand variants of SIINFEKL were topically applied to the skin 1 day after recruitment to the skin. (**B**) Representative images and (**C**) quantification of epidermal whole mounts (CD90.1 cyan) of VV-OVA and DNFB+ APL treated flanks, at steady state (no recall) or 6 days post-recall response. Each symbol represents data from an individual animal. Data are representative of 5 independent experiments. *$p<0.05$ by unpaired Student's t-tests. Scale bar, 50 µm. Panel A created with BioRender.com.

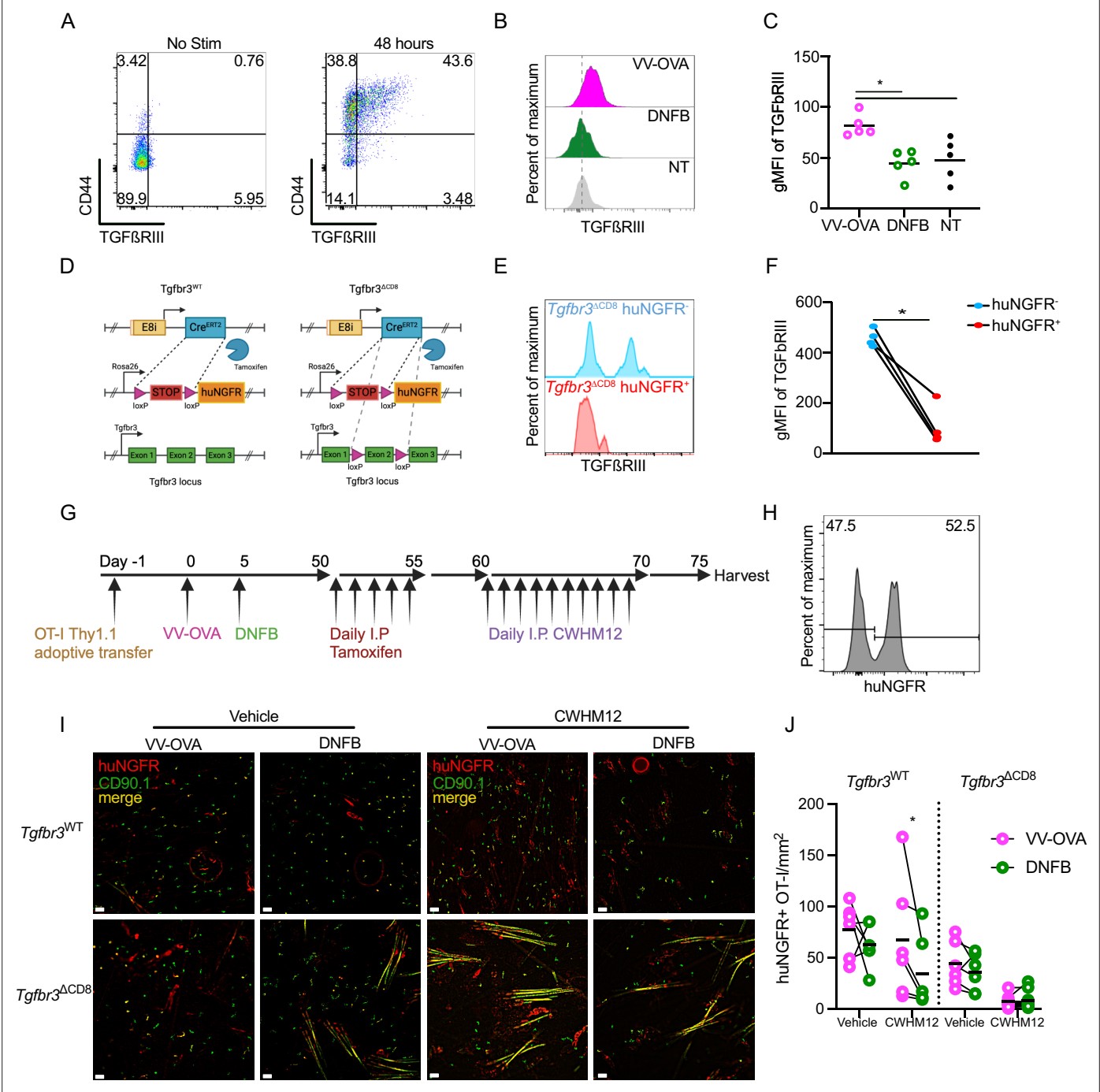

**Figure 6.** Local *antigen experienced T_RM have improved persistence mediated by TGFßRIII*. (**A**) Representative flow cytometric plots of TGFßRIII staining of Thy1.1+ OT-I cells stimulated in vitro for 48 hr with PBS or anti-CD3 anti-CD28. (**B**) Representative histograms showing TGFßRIII expression in CD45+ CD3+ CD8+ CD90.1+ gated OT-I cells isolated from VV-OVA, DNFB or untreated flanks at least 50 days post infection. (**C**) Quantification of (**B**). (**D**) Schematic demonstrating genetics of Tgfbr3^WT and Tgfbr3^ΔCD8 mice. (**E**) Representative histogram and (**F**) quantification of TGFßRIII expression in huNGFR- or huNGFR+ Tgfbr3^ΔCD8 T cells harvested 5 days following i.p. treatment with tamoxifen then stimulated in vitro for 48 hr with anti-CD3 and anti-CD28. (**G**) Experimental scheme. (**H**) Representative histogram of CD45.2+CD3+CD8+CD90.1+OT I cells isolated from LNs after tamoxifen treatment demonstrating transformation efficiency. (**I**) Representative epidermal whole mounts showing Thy1.1 staining (green), huNGFR staining (red) or merge (yellow) of VV-OVA or DNFB treated flanks from mice adoptively transferred with either Tgfbr3^WT or Tgfbr3^ΔCD8 cells, treated with tamoxifen i.p., and then given either PBS or i.p. CWHM12 for 10 days. Hair follicles in the sample present as long yellow streaks. (**J**) Quantification of total huNGFR+ Thy1.1+ OT-I in (**I**). Each symbol represents data from an individual animal. Black lines represent group means. Data is representative of 3 independent experiments. *P<0.05 by Dunnett's test (**C**) or paired Student's t-tests (**F**) and (**J**). Unpaired Student's t-tests between Tgfbr3^WT and Tgfbr3^ΔCD8 vehicle-treated groups show non-significance for both VV-OVA and DNFB treated flanks. Scale bar represents 50 μm. Panel D and G were created with BioRender.com.

To ablate TGFβRIII from $T_{RM}$ once they have fully differentiated in the epidermis, we next adoptively transferred CD8[+] T cells isolated from either Tgfbr3[WT] or Tgfbr3[ΔCD8] mice into naïve WT mice followed by skin VV-OVA infection on the left flank. On day 5 post-infection, the right flank was treated with DNFB. After at least 50 days of rest to allow for $T_{RM}$ formation, both cohorts were treated daily with 0.05 mg/g tamoxifen i.p. to transform approximately 50% of adoptively transferred cells, generating populations of both control huNGFR negative and huNGFR positive OT-I cells. To test the effects of TGFβRIII ablation, TGFβ activation was inhibited for 10 days by i.p. administration of CWHM12, a small molecule inhibitor of integrins αvβ6 and αvβ8 (**Figure 6G–H**). Analysis of epidermal whole mounts by immunofluorescence revealed that blockade of TGFβ activation reduced the number of bystander Tgfbr3[WT] $T_{RM}$ at the DNFB site, but not local antigen experienced $T_{RM}$ at the VV-OVA site, consistent with our earlier results (**Figure 6I–J**). In contrast, local antigen experienced huNGFR[+] Tgfbr3[ΔCD8] $T_{RM}$ from the VV-OVA site that lack TGFβRIII were efficiently depleted from the epidermis by CWHM12 treatment. The depletion of bystander huNGFR[+] Tgfbr3[ΔCD8] $T_{RM}$ at the DNFB site was augmented. Notably, numbers of Tgfbr3[ΔCD8] $T_{RM}$ in cohorts treated with vehicle did not induce a statistically significant reduction in steady-state $T_{RM}$, indicating that loss of TGFβRIII is not an absolute requirement for $T_{RM}$ epidermal residence in the steady state. Thus, expression of TGFβRIII by local antigen experienced $T_{RM}$ which is downstream of TCR ligation is required for their capacity to remain in the epidermis when activated TGFβ is limiting.

## Discussion

Herein, we have demonstrated that the increased capacity of local antigen experienced $T_{RM}$ to persist in the epidermis when levels of TGFβ are limited is mediated by increased expression of TGFβRIII. We also show that local antigen experienced $T_{RM}$ have increased proliferative capacity during repeated antigen recalls. In addition, the increased proliferative capacity was directly correlated with the strength of TCR stimulation during $T_{RM}$ development. Finally, we found that local antigen experienced $T_{RM}$ appear more transcriptionally related to fully differentiated $T_{RM}$. Taken together, these data support a model in which TCR engagement by cognate antigen in the skin is a required final step in $T_{RM}$ differentiation resulting in their increased fitness exemplified by increased proliferative capacity and the ability to persist in the epidermis when active TGFβ is limited.

We propose that the augmented fitness of local antigen experienced $T_{RM}$ represents a mechanism to enrich for high avidity TCR clones in the epidermis. Skin inflammation recruits $T_{EFF}$ into the skin, some of which develop into $T_{RM}$. In the absence of competition with pre-existing $T_{RM}$, $T_{RM}$ form comparably in the presence or absence of cognate antigen. Thus, we found equivalent numbers of bystander $T_{RM}$ at the DNFB and local antigen experienced $T_{RM}$ at the VV sites with both TCR transgenic and endogenous T cells. In contrast, when new $T_{EFF}$ are recruited into sites with pre-existing $T_{RM}$, there is clonal competition for limited amounts of active TGFβ, resulting in enrichment of fitter, local antigen experienced $T_{RM}$ (**Hirai et al., 2021**). We now find that this enrichment likely results from 2 different competitive advantages. First, antigen encounter in the skin results in increased expression of TGFβRIII, which increases TGFβ avidity for the signaling by the TGFβ receptor. Since TGFβ signaling is required for epidermal persistence, this would provide an advantage for local antigen experienced $T_{RM}$ over bystanders. Second, local antigen experienced $T_{RM}$ have increased proliferation when re-encountering antigen in the epidermis. Following repeated challenges which would be expected outside of SPF conditions, the combination of improved expansion and persistence would work together to enrich for high avidity clones, thereby shaping the epidermal CD8[+] T cell memory pool. Recently, it has been observed that $T_{RM}$ can contribute significantly to the pool of circulating memory cells (**Steinert et al., 2015**; **Beura et al., 2018b**; **Fonseca et al., 2020**; **Wijeyesinghe et al., 2021**; **Behr et al., 2020**). Thus, mechanisms augmenting epidermal $T_{RM}$ fitness that shape the pool of epidermal $T_{RM}$ may also affect the pool of systemic memory cells and represent an example of extra-thymic clonal section.

When $T_{RM}$ were challenged in a primary antigen recall response, we noted that local antigen experienced $T_{RM}$ expanded to a greater extent than bystanders. This expansion resulted from increased in-situ proliferation with minimal contribution from newly recruited $T_{EFF}$, consistent with prior reports (**Park et al., 2018**; **Beura et al., 2018a**; **Çuburu et al., 2012**). Interestingly, a similar phenomenon occurred following a second encounter with antigen. This indicates that an encounter with peptide at a late time point after $T_{RM}$ differentiation (>50 days) is insufficient to convert bystander $T_{RM}$ into local

antigen experienced T$_{RM}$. Thus, there appears to be a window during T$_{RM}$ development when TCR engagement can allow for full differentiation. We also observed after a single recall response that T$_{RM}$ contracted to an elevated baseline, suggesting an increase in the epidermal niche. We speculate this may result from a reduced T cell intrinsic requirement for survival and/or homeostatic proliferation factors, such as IL-7 or IL-15 or increased expression of these factors by keratinocytes (*Richmond et al., 2018*; *Adachi et al., 2015*). Altered sensitivity or availability of TGFβ is unlikely to explain the increased niche size, as this would be predicted to vary between local antigen experienced and bystander.

Transcriptional analysis of T$_{RM}$ isolated from the small intestine have revealed intra-organ heterogeneity, with unique transcriptional populations arising early during T$_{RM}$ development (*Milner et al., 2020*; *Kurd et al., 2020*; *Fitz Patrick et al., 2021*). This aligns well with our identification of 6 distinct transcriptional clusters of epidermal T$_{RM}$. Cluster 3 appears to represent fully differentiated T$_{RM}$ based on comparison with other T$_{RM}$ datasets. In addition, cluster 3 cells more highly expressed the activation and proliferation-associated genes *Junb, Fos* and *Dusp1* as well as *Nr4a1*. Increased basal expression of the AP-1 family members *Junb and Fos* could contribute to the enhanced proliferation of antigen-experienced epidermal T$_{RM}$ during a recall response. Intriguingly, memory CD8 T cells lacking the transcription factor Zbtb20 manifest elevated expression of AP-1 family members and mount more robust antitumor responses (*Hao et al., 2024*). The *Nr4a1* gene encodes for Nur77, which is induced by TCR signaling and its expression correlates with peptide avidity. Notably, Nur77 is required for T$_{RM}$ formation in the liver (*Mackay et al., 2013*; *Mackay et al., 2015*; *Aluwihare et al., 2009*; *Jennings et al., 2020*; *Boddupalli et al., 2016*). Interestingly, cells in cluster 3 only accounted for 27% of T$_{RM}$ that had the opportunity to encounter their cognate antigen in the VV-treated flank. We speculate that not all clones at the VV site fully develop into fitter T$_{RM}$ due to lower TCR avidity or specificity to viral antigens only expressed early during infection, which would be absent once the clones arrived into skin.

In sum, TCR signaling during T$_{RM}$ differentiation represents a previously unappreciated final step in T$_{RM}$ differentiation. This results in fitter T$_{RM}$ with a lower requirement for TGFβ transactivation due to increased expression of TGFβRIII and enhanced proliferation in response to peptide stimulation. Moreover, the differing responses to altered peptide ligands indicate that the degree of fitness depends on TCR signal strength. Thus, polyclonal T$_{RM}$ likely develop into a spectrum of bystander to local antigen experienced cells based on TCR avidity. Though we have focused entirely on epidermal T cells, we suspect that these mechanisms may play a role in other epithelial tissues where residency is also dependent upon TGFβ. Additionally, we have solely investigated memory CD8$^+$ T cells after acute inflammation; the role of ongoing TCR-engagement during chronic antigen encounter remains unexplored.

# Materials and methods

## Mice

We generated Thy1.1$^+$*Rag1*$^{-/-}$OT-I mice by crossing OT-I mice with *Rag1*$^{-/-}$ and Thy1.1 mice. E8I-*cre*ER$^{T2}$ and ROSA26.LSL.hNGFR reporter mice were developed by Dario A.A. Vignali (University of Pittsburgh) (*Hirai et al., 2019*). E8I-*cre*ER$^{T2}$ mice and Thy1.1$^+$*Rag1*$^{-/-}$OT-I mice were crossed with ROSA26.LSL.hNGFR reporter mice to generate Tgfbr3$^{WT}$ mice and additionally with *Tgfbr3*$^{fl/fl}$ mice to obtain Tgfbr3$^{\Delta CD8}$ mice. *Tgfbr3*$^{fl/fl}$ mice were developed by Herbert Y Lin (Program in Membrane Biology/Nephrology, Department of Medicine, Massachusetts General Hospital and Harvard Medical School, Boston, Massachusetts) (*Li et al., 2018*). We used age- and sex-matched female mice that were between 6 and 12 weeks of age at the start of all experiments. All mice were maintained under specific-pathogen-free conditions and all animal experiments were approved by the University of Pittsburgh Institutional Animal Care and Use Committee under the Animal Welfare Assurance Number D16-00118 (A3187-01).

## T$_{RM}$ cell models and blocking TGFβ activation treatments

Mice were infected by skin scarification (skin infection) with $3 \times 10^6$ plaque-forming units recombinant vaccinia virus expressing the SIINFEKL peptide of ovalbumin (VV-OVA), or vaccinia virus without recombinant peptide expression (VV). For skin scarification, 45 µL of VV-OVA or VV was applied to shaved left flank (4–5 cm$^2$) and the skin were gently scratched 100 times with 27 G needle under

anesthesia. VV or VV-OVA infected mice were further treated with 0.15% 1-Fluoro-2,4-dinitrobenzene (DNFB, D1529: Sigma-Aldrich) in 4:1 acetone: olive oil (O1514, Sigma-Aldrich) on the flank opposite the site of infection (40 μL) 5 days post infection. In some experiments, 4:1 acetone and olive oil was added to the stock solution of OVA$_{257–264}$ (SIINFEKL or SIYNFEKL, RP101611, Genscript/Fisher) in DMSO (D2650, Sigma-Aldrich) (10 mg/mL) for a final concentration of OVA$_{257–264}$ at 0.2 mg/mL and 50 μL was painted to the DNFB-treated skin 1 day after DNFB treatment.

### Recall response

50+days after VV-OVA infection. 4:1 acetone and olive oil was added to the stock solution of OVA$_{257–264}$ (SIINFEKL, RP101611, Genscript/Fisher) in DMSO (D2650, Sigma-Aldrich) (10 mg/mL) for a final concentration of OVA$_{257–264}$ at 0.2 mg/mL and 50 μL was painted to both flanks.

### CWHM12 treatments

For blocking TGFβ activation with a small molecule integrin inhibitor, compound CWHM12 (kindly provided by Indalo Therapeutics, Cambridge, MA) was solubilized in 100% DMSO. Further dilution to 50% DMSO was made in sterile 100% PBS and dosed to 100 mg per kg body weight per day delivered by i.p. injections once a day for 10 days.

### Adoptive transfers

OT-I cells were purified from spleen and lymph nodes by MojoSort Mouse CD8 T Cell Isolation Kit (48007, BioLegend) according to the manufacturer's instructions and $1 \times 10^5$ OT-I cells were intravenously transferred in all experiments. All mice were allowed to rest for 1 day before further experimentation.

### Tamoxifen treatment

Tamoxifen (T5648; Sigma-Aldrich) was dissolved in 1/10th volume of 200 proof ethanol following incubation at 37°C for 15–30 min with 300 rpm shaking. Corn oil (C8267, Sigma-Aldrich) was added for a final concentration of Tamoxifen at 10 mg/ml and was administered to mice for 5 consecutive days by intraperitoneal injection at 0.05 mg per g body weight, in order to transform approximately 50% of adoptively transferred cells.

### BrdU treatment

BrdU (B5002, Sigma-Aldrich) was dissolved in sterile PBS. 2 mg in 200 uL of PBS was injected i.p. daily into each mouse for 2 consecutive days before harvest.

### Immunofluorescence of epidermis

Epidermal sheets were prepared as previously described (*Mohammed et al., 2016*). Briefly, 2 square cm sections were harvested 2 cm distal from the mid-clavicular line for consistency, and then the epidermal side of shaved defatted flank skin was affixed to slides with double-sided adhesive (3M, St. Paul, MN). Slides were incubated in 10 mM EDTA in PBS for 90 min at 37°C, followed by physical removal of the dermis. The epidermal sheets were fixed in 4% PFA at RT for 15 min. The epidermal sheets were blocked with PBS containing 0.1% Tween-20, 2% BSA and 2% rat serum for 1 hr at RT before staining for 1 hour with antibodies at RT in PBS containing 0.1% Tween-20 and 0.5% BSA. The slides were mounted with ProLong Gold Antifade Mount with DNA stains DAPI (P36931, Thermofisher). Images were captured on a Keyence BZX800 fluorescent microscope (Keyence, Osaka, Japan). Analysis was performed using BZ-H4A Advanced Analysis Software (Keyence, Osaka, Japan). For the enumeration of cells, three images from distant sites randomly determined within an epidermal sheet from a mouse were counted (total 3 mm$^2$) manually after image processing by Adobe Photoshop (version 6) and the average number per mm$^2$ epidermis was calculated as representative of the epidermal sheet. Anti-CD8α (53–6.7), Thy1.1 (OX-7) and huNGFR (ME20.4) were purchased from BioLegend.

### Depletion of circulation cells

For depletion of circulating memory OT-I cells, OT-I adoptive transfer VV-OVA and DNFB treated mice were injected i.p. with 0.3–1 μg anti-Thy1.1 (HIS51, ThermoFisher) in 200 μL PBS for 10 consecutive

days. Depletion of OT-I T cells (<0.5% of total CD8$^+$ T cells and <1%) were confirmed by staining with anti-Thy1.1 (OX-7) using blood 3 days after depletion. For FTY720 treatment, OT-I adoptive transfer VV-OVA and DNFB-treated mice were injected i.p. with FTY720 (10006292, Fisher/Cayan Chemical) at 1 µg/g body weight in sterile PBS with 0.5% DMSO (D2650, Sigma-Aldrich) daily for 6 days. Depletion of circulating cells was confirmed with staining of anti-Thy1.1 (OX-7) and anti-CD8α (53–6.7) using blood 3 days after the end of treatment.

## Flow cytometry

Preparation of single cell suspension from skin, shaved skin was harvested, and fat tissues was removed by forceps mechanically. The skin was then mechanically digested with either scissors or a gentleMACS Dissociator (Miltenyi Biotec, Bergisch Gladbach, Germany), and then resuspended in RPMI 1640 media (Gibco, Grand Island, NY) containing 2.5 mg/ml Collagenase XI (9001-12-1, Sigma-Aldrich), 0.1 mg/ml DNase (04536282001, Sigma-Aldrich), 0.01 M 4-(2-hydroxyethyl)-1-piperazinee thanesulfonic acid (7365-45-9, Sigma-Aldrich), and 10% fetal bovine serum, followed by incubation in a shaking incubator for 30 min at 37°C. The resulting cell mesh was filtered through a 40-µm cell strainer (BD Biosciences, San Jose, CA). The single-cell suspension was then resuspended in 44% Percoll (89428-524, VWR) in RPMI and layered over 67% Percoll in PBS. After 20 min of centrifugation at 931 x g at RT, the interface was isolated and resuspended in FACS buffer. LNs (axillary and inguinal) were incubated in 400 U/ml Collagenase D (Roche Applied Science, Penzberg, Germany) and 0.1 mg/ml DNase in RPMI 1640 with 10% fetal bovine serum for 40 min at 37°C and then minced through a 40-µm cell strainer. In some experiments, LNs were heat-treated at 60°C in a hot water bath for 1 hour. Blood was collected with heparin (H3393, Sigma-Aldrich) and treated with red blood cell lysing buffer (R7757, Sigma-Aldrich). Single-cell suspensions were blocked with 2.4G2 culture supernatant (ATCC, Manassas, VA). Surface staining was performed in standard FACS buffer for 30 min at 4°C. Anti-CD8α (53–6.7), CD44 (IM7), CD3 (17A2), Thy1.2 (30-H12), Thy1.1 (OX-7), CD69 (H1.2F3), CD103 (2E7), BrdU (3D4), MHC-II (M5/114.15.2)**,** CD45.2 (104), and Annexin V were purchased from BioLegend. Anti-TGFßRIII (1C5H11) was purchased from Sigma-Aldrich. Soluble tetrameric B8R$_{20–27}$/H2-K$^b$ complex (made by the NIH Tetramer Core Facility) was conjugated to PE-labeled streptavidin. A BD LSRFORTESSA (BD Biosciences) and FlowJo software (TreeStar, Ashland, OR) were used for analysis.

## Single-cell RNA and totalseq C-sequencing - sample preparation

### Flow cytometry and HT antibody staining

Single-cell lymphocyte suspensions were isolated as described above and stained with anti-CD45-APC, anti-CD8a, anti-CD44, anti-CD103 and different TotalSeq-C Anti-Mouse Hashtags (TotalSeq-C Anti-Mouse Hashtags 1–10 BioLegend #155861–155879, in staining buffer (DMEM, 2% FCS and 10 mM HEPES)) for 20 min at 4°C in the dark. Standard spleen cell suspensions were stained under identical conditions but included anti-CD45-FITC (BioLegend 109806, clone 104) to differentiate them from other samples when pooled. Each sample was labeled with a unique hashtag, enabling the downstream assignment of individual cell (10x cell barcode) to their respective source samples (*Stoeckius et al., 2018*).

## Cell sorting

For each sample, 500–5000K DAPI$^-$ CD45$^+$ Thy1.2$^+$ CD8a$^+$ (skin samples) or DAPI$^-$ CD45$^+$ Thy1.2$^+$ CD8a$^+$ CD44$^+$ CD103$^+$ (LN samples) T cells were sorted on the FACSAria III and pooled in a single collection tube at 4C. DAPI was added just before the sort.

## ImmGenT TotalSeq-C custom mouse panel staining

The pooled single-cell suspension was stained with the ImmGenT TotalSeq-C custom mouse panel, containing 128 antibodies (BioLegend Part no. 900004815) and FcBlock (Bio X Cell #BE0307). Because 500,000 cells are required for staining, unstained splenocytes were spiked in to reach a total of 500,000 cells.

## Second sort

Cells were subsequently sorted a second time with the addition of DAPI to select for live T cells, and include 5000 total splenocyte standard cells that were sorted into a single collection tube.

## Single-cell RNA and totalseq C-sequencing - library preparation

### Cell encapsulation and cDNA library

Single-cell RNA sequencing was performed using the 10x Genomics 5' v2 platform with Feature Barcoding for Cell Surface Protein and Immune Receptor Mapping, adhering to the manufacturer's guidelines (CG000330). After cell encapsulation with the Chromium Controller, reverse transcription and PCR amplification were performed in the emulsion. From the amplified cDNA library, smaller fragments containing TotalSeq-C-derived cDNA were separated for Feature Barcode library construction, while larger fragments containing transcript-derived cDNA were preserved for TCR and Gene Expression library generation. Library sizes for both cDNA fractions were evaluated using the Agilent Bioanalyzer 2100 High Sensitivity DNA assay and quantified with a Qubit dsDNA HS Assay kit on a Qubit 4.0 Fluorometer.

### RNA library construction

After enzymatic fragmentation and size selection of the cDNA, the library was ligated to an Illumina R2 sequence and indexed using unique Dual Index TT set A index sequences, SI-TT-B6.

### TotalSeq-C library construction

Totalseq-C derived cDNA was processed into the Feature Barcode libraries following the manufacturer's protocol. The library was indexed with a unique dual index TN set TN set A (10x part no. 3000510) index sequence SI-TN-F9.

### Sequencing

The three libraries were pooled based on molarity in the following proportions: 47.5% RNA, 47.5% Feature Barcode, and 5% TCR. The pooled libraries were sequenced on an Illumina NovaSeq S2 platform (100 cycles) using the 10x Genomics specifications: 26 cycles for Read 1, 10 cycles for Index 1, 10 cycles for Index 2, and 90 cycles for Read 2.

## Single-cell RNA, TCR and totalseq C-sequencing - data processing

### Code

Code is available on https://github.com/dzemmour/immgen_t, copy archived at *Casey and Zemmour, 2025*.

### Count matrices

Gene and TotalSeq-C antibody (surface protein panel and hashtags) counts were obtained by aligning reads to the mm10 (GRCm38) mouse genome using the M25 (GRCm38.p6) Gencode annotation and the DNA barcodes for the TotalSeq-C panel. Alignment was performed with CellRanger software (v7.1.0, 10x Genomics) using default parameters. Cells were identified and separated from droplets with high RNA and TotalSeq-C counts by determining inflection points on the total count curve, using the barcodeRanks function from the DropletUtils package.

### Sample demultiplexing

Sample demultiplexing was performed using hashtag counts and the HTODemux function from the Seurat package (Seurat v4.1). Doublets (droplets containing two hashtags) were excluded, and cells were assigned to the hashtag with the highest signal, provided it had at least 10 counts and was more than double the signal of the second most abundant hashtag. Hashtag count data were also visualized using t-SNE to ensure clear separation of clusters corresponding to each hashtag. All single cells from the gene count matrix were uniquely matched to a single hashtag, thereby linking them unambiguously to their original sample.

### Quality control (QC) and batch correction

Cells meeting any of the following criteria were excluded from the analysis: fewer than 500 RNA counts, dead cells with over 10% of counts mapping to mitochondrial genes, fewer than 500 TotalSeq-C counts, or positivity for two isotype controls (indicating non-specific TotalSeq-C antibody

binding). Non-T cells were excluded based on the expression of the MNP gene signature, B cell signature, ILC gene signature, and the absence of T cell gene signature (score calculated using AddModuleScore_UCell). CITE-seq data did not meet quality control and was not used in the analysis.

## ImmGen T integration

The data was integrated with the rest of the ImmgenT dataset using the SCVI.TOTALVI model (v1.2.0) and the 10x lane as a batch parameter. Dimensionality reduction was performed using the pymde. preserve_neighbors() function with default parameters [https://pymde.org/citing/index.html]. Cell clustering was carried out using the FindClusters() function in Seurat. Manual annotation by the immgenT consortium was done using protein and RNA expression of Cd3e, Trbc1, Trbc2, Cd4, Cd8a, Cd8b1, Foxp3, Mki67, Sell, Cd44, Trgc1, Itgax, Itgam, Ms4a1, CD3, TCRB, THY1.2, CD4, CD8A, CD8B, CD62L, CD44, TCRGD, TCRVG1.1, TCRVG2, TCRVG3, CD19, CD20, ITAM.CD11B, ITAX. CD11C, KLRBC-NK1.1. Data was visualized using the Rosetta software (https://cbdm.connect.hms. harvard.edu/ImmgenT/PublicRosetta/). ImmgenT integration and annotation available on https:// www.immgen.org/ImmGenT/.

The data discussed in this publication have been deposited in NCBI's Gene Expression Omnibus (*Hao et al., 2024*). The SubSeries data discussed in this paper are accessible through the GEO accession number GSE283941 (https://www.ncbi.nlm.nih.gov/geo/query/acc.cgi?acc=GSE283941). The ImmgenT SuperSeries data are accessible through the GEO accession number GSE297097 (https:// www.ncbi.nlm.nih.gov/geo/query/acc.cgi?acc=GSE297097).

## Clustering and dimensionality reduction

Using Seurat v4.2. [https://pubmed.ncbi.nlm.nih.gov/31178118/], the variance-stabilizing transformation (VST method) was applied, and PCA was conducted on the top 2000 genes. Principal components explaining 80% of the total variance were selected for two-dimensional reduction using UMAP. Clustering was performed on these principal components using the FindClusters() function in Seurat.

## Differential Expression

To determine differentially expressed genes, FindMarkers() within Seurat was used. AddModuleScore() was used to visualize aggregated expression of a set of genes. Volcano plots were created using EnhancedVolcano (v1.14.0).

## Signature score analysis

Signature score was calculated by first generating differentially expressed genes of each cluster or condition compared to naïve T cells from the spleen control, and then calculating an average z-score for every up and down gene in a core gene signature datasets. The core gene datasets were generated by *Jaiswal et al., 2022* by calculating the exclusive expression of genes from T-cells in the datasets GSE47045 (*Mackay et al., 2013*), GSE10239 (*Sarkar et al., 2008*) and GSE41867 (*Doering et al., 2012*). Comparative programs for $T_{RM}$ development over time were analyzed against the transcripts taken from the top 100 differentially expressed genes of each $T_{RM}$ timepoint/ naïve from GSE79805 (*Pan et al., 2017*) by adapting published scoring methods with this data set (*Jaiswal et al., 2022*). The overall signature score was scored based on expression data that was quantified using RSEM in transcripts per million (TPM), then log-transformed as log2 (TPM+ 1). It was then centered for each gene across all cells. The centered data was averaged across sets of genes to define signature scores. We subtracted a control score from the signature score, which is defined using the same process on randomly selected gene sets. A randomized control was generated by the average enrichment of each group compared to a randomly generated gene set of an equal number of probes, as iterated (*Jaiswal et al., 2022*).

## Quantification and statistical analysis

Groups were compared with Prism software (GraphPad) using two-tailed paired Student's t-test for comparison of left and right flanks, two-tailed unpaired Student's t-test for comparison between animals, or Dunnett's test for comparisons of more than two groups. Sample size was determined by an a priori power analysis with an α of 0.05 and a 1-β of 0.80 based on pilot data. Data is presented as

each data point and mean with the standard error of the mean (s.e.m.). *P*<0.05 was considered significant. Studies were designed in accordance with ARRIVE guidelines. Researchers were blinded for all sample analysis. All experimental data was included in the figures, and mice were randomly assigned to groups when appropriate.

## Acknowledgements

We thank Daniel Bernard (Division of Endocrinology and Metabolism, McGill University, Montreal, Canada) for providing the Tgfbr3$^{fl/fl}$ mice. We thank the members of the Kaplan and Vignali laboratories and members throughout the departments of Dermatology and Immunology for helpful discussions. We also thank the Division of Laboratory Animal Resources of the University of Pittsburgh for excellent animal care.

## Additional information

### Competing interests

Niroshana Anandasabapathy: serves as a consultant or is on the advisory board for Shennon Biotechnologies, Panther Life Sciences, Verrica pharmaceuticals, Genmab, 23 and me, Johnson and Johnson, Lytix Biopharma. Daniel H Kaplan: is a paid consultant for AbbVie Inc, Beiersdorf AG, Janssen Research and Development LLC, and Aditum Bio; has a sponsored research agreement with Galderma Laboratories, Lp. The other authors declare that no competing interests exist.

### Funding

| Funder | Grant reference number | Author |
|---|---|---|
| National Institute of Arthritis and Musculoskeletal and Skin Diseases | AR083713 | Daniel H Kaplan |
| National Institutes of Health | 2T32AI060525 | Eric S Weiss |
| National Institutes of Health | 5T32AI089443 | Eric S Weiss |
| National Institutes of Health | 5R01 AR083208 | Niroshana Anandasabapathy |
| National Institutes of Health | 5R01AR060744 | Daniel H Kaplan |

The funders had no role in study design, data collection and interpretation, or the decision to submit the work for publication.

### Author contributions

Eric S Weiss, Conceptualization, Formal analysis, Investigation, Methodology, Writing – original draft, Writing – review and editing; Toshiro Hirai, Conceptualization, Formal analysis, Investigation, Methodology; Haiyue Li, Andrew Liu, Shannon Baker, Jacob Gillis, Youran R Zhang, Torben Ramcke, Kazuo Kurihara, Investigation; Ian Magill, Data curation, Formal analysis, Investigation; The ImmGen Consortium OpenSource T cell Project, Data curation, Formal analysis; David Masopust, Conceptualization; Niroshana Anandasabapathy, Software, Methodology; Harinder Singh, Conceptualization, Funding acquisition, Methodology, Writing – review and editing; David Zemmour, Resources, Data curation, Software, Formal analysis, Visualization, Methodology; Laura K Mackay, Investigation, Methodology; Daniel H Kaplan, Conceptualization, Resources, Supervision, Funding acquisition, Project administration, Writing – review and editing

### Author ORCIDs

Eric S Weiss ⓘ https://orcid.org/0000-0003-2352-0036
Daniel H Kaplan ⓘ https://orcid.org/0000-0003-0598-0047

### Ethics

All mice were maintained under specific-pathogen-free conditions and all animal experiments were approved by University of Pittsburgh Institutional Animal Care and Use Committee under the Animal Welfare Assurance Number D16-00118 (A3187-01).

Reviewer #1 (Public review): https://doi.org/10.7554/eLife.107096.3.sa1
Reviewer #2 (Public review): https://doi.org/10.7554/eLife.107096.3.sa2
Author response https://doi.org/10.7554/eLife.107096.3.sa3

## Additional files

### Supplementary files

MDAR checklist

### Data availability

All scRNA-seq is available within GEO GSE283941 and GSE297097. Code used in this paper can be found at https://www.immgen.org/ImmGenT/, or it is cited within the text.

The following datasets were generated:

| Author(s) | Year | Dataset title | Dataset URL | Database and Identifier |
|---|---|---|---|---|
| Weiss ES, Hirai T, Li H, Lui A, Baker S, Magill I, Fan J, Masopust D, Anandasabapathy N, Singh H, Zemmour D, MacKay L, Kaplan DH | 2024 | Development of Fit Epidermal Resident Memory T Cells Requires Antigen Encounter in the Skin | https://www.ncbi.nlm.nih.gov/geo/query/acc.cgi?acc=GSE283941 | NCBI Gene Expression Omnibus, GSE283941 |
| Zemmour D, Goldrath A, Kronenberg M, Kang J, Benoist C | 2025 | The ImmgenT Open-Source project | https://www.ncbi.nlm.nih.gov/geo/query/acc.cgi?acc=GSE297097 | NCBI Gene Expression Omnibus, GSE297097 |

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

# Appendix 1

## Appendix 1—key resources table

| Reagent type (species) or resource | Designation | Source or reference | Identifiers | Additional information |
|---|---|---|---|---|
| Antibody | Brilliant Violet 605 anti-mouse CD8a (Rat monoclonal, 53–6.7) | BioLegend | Cat# 100744, RRID:AB_2562609 | 1:200 |
| Antibody | Alexa Fluor 647 anti-mouse CD90.1 (mouse monoclonal, OX-7) | BioLegend | Cat# 202508, RRID:AB_492884 | 1:200 |
| Antibody | Alexa Fluor 647 anti-human CD271 (NGFR) (mouse monoclonal, ME20.4) | BioLegend | Cat# 345114, RRID:AB_2572059 | 1:200 |
| Antibody | Anti-mouse Thy1.1 (mouse monoclonal, HIS51) | Thermofisher | Cat# 14-0900-85, RRID:AB_467374 | 1:100 |
| Antibody | PerCP/Cyanine5.5 anti-mouse/human CD44 (rat monoclonal, IM7) | BioLegend | Cat# 103032, RRID:AB_2076204 | 1:200 |
| Antibody | Alexa Fluor 700 anti-mouse CD3 (rat monoclonal, 17A2) | BioLegend | Cat# 100216, RRID:AB_493697 | 1:200 |
| Antibody | FITC anti-mouse CD90.2 (rat monoclonal, 30-H12) | BioLegend | Cat# 105305, RRID:AB_313176 | 1:200 |
| Antibody | BUV737 anti-mouse CD69 (Armenian hamster monoclonal, H1.2F3) | BD Biosciences | Cat# 612793, RRID:AB_2870120 | 1:200 |
| Antibody | Brilliant Violet 510 anti-mouse CD103 (Armenian hamster monoclonal, 2E7) | BioLegend | Cat# 121423, RRID:AB_2562713 | 1:200 |
| Antibody | FITC anti-BrdU (mouse monoclonal, 3D4) | BioLegend | Cat# 364103, RRID:AB_2564480 | 1:200 |
| Antibody | APC anti-mouse CD45.2 (mouse monoclonal, 104) | BioLegend | Cat# 109841, RRID:AB_2563485 | 1:200 |
| Antibody | FITC anti-mouse CD45.2 (mouse monoclonal, 104) | BioLegend | Cat# 109805, RRID:AB_313442 | 1:200 |
| Antibody | Alexa Fluor 700 anti-mouse I-A/I-E (rat monoclonal, M5/144.15.2) | BioLegend | Cat# 107621, RRID:AB_493726 | 1:200 |
| Antibody | Anti-TGFβRIII (1C5H11) | Novus | Cat# NBP2-37418, RRID:AB_3296314 | 1:200 |
| Chemical compound, drug | DNFB | Sigma-Aldrich | D1529 | 1-fluoro-2,4-dinitrobenzene |
| Chemical compound, drug | Olive oil | Sigma-Aldrich | O1514 | |
| Chemical compound, drug | DMSO | Sigma-Aldrich | D2650 | Dimethyl sulfoxide |
| Chemical compound, drug | CWHM12 | Indalo therapeutics | N/A | |
| Chemical compound, drug | Tamoxifen | Sigma-Aldrich | T5648 | |
| Chemical compound, drug | Corn oil | Sigma-Aldrich | C8267 | |
| Chemical compound, drug | BrdU | Sigma-Aldrich | B5002 | |
| Chemical compound, drug | ProLong Gold Antifade Mount with DNA stains DAPI | Thermofisher | P36931 | |

*Appendix 1 Continued on next page*

*Appendix 1 Continued*

| Reagent type (species) or resource | Designation | Source or reference | Identifiers | Additional information |
|---|---|---|---|---|
| Chemical compound, drug | FTY720 | Fisher/Cayan Chemical | Cat# 10006292 | |
| Chemical compound, drug | Collagenase XI | Sigma-Aldrich | Cat# 9001-12-1 | |
| Chemical compound, drug | DNase | Sigma-Aldrich | Cat# 04536282001 | |
| Chemical compound, drug | 4-(2-hydroxyethyl)–1-piperazineethanesulfonic acid | Sigma-Aldrich | Cat# 7365-45-9 | |
| Chemical compound, drug | Percoll | VWR | Cat# 89428–524 | |
| Chemical compound, drug | Collagenase D | Sigma-Aldrich | Cat# 11088866001 | |
| Chemical compound, drug | Heparin | Sigma-Aldrich | Cat# H3393 | |
| Chemical compound, drug | Red blood cell lysing buffer | Sigma-Aldrich | Cat# R7757 | |
| Chemical compound, drug | 2.4G2 culture supernatant | ATCC | HB-197 | |
| Commercial assay or kit | MojoSort Mouse CD8 T Cell Isolation Kit | BioLegend | 48007 | |
| Commercial assay or kit | Chromium Next GEM Single Cell 5' Reagent Kits V2 | 10x Genomics | Cat# CG000330 | |
| Commercial assay or kit | dual index TN set TN set A | 10x Genomics | part no. 3000510 | |
| Gene (*Mus musculus*) | Thy1.1 | The Jackson Laboratory | Jax stock #005443, CBy.PL(B6)-Thy1a/ScrJ | |
| Gene (*Mus musculus*) | OT-I | The Jackson Laboratory | Jax stock #003831, C57BL/6-Tg(TcraTcrb)1100Mjb/J | |
| Gene (*Mus musculus*) | *Rag1*$^{-/-}$ | The Jackson Laboratory | Jax stock #002216, Rag1-/-: B6.129S7-Rag1tm1Mom/J | |
| Gene (*Mus musculus*) | E8I-*cre*ER$^{T2}$ | Dario A.A. Vignali (University of Pittsburgh); *Hirai et al., 2019* | | |
| Gene (*Mus musculus*) | ROSA26.LSL.hNGFR | Dario A.A. Vignali (University of Pittsburgh); *Hirai et al., 2019* | | |
| Gene (*Mus musculus*) | *Tgfbr3*$^{fl/fl}$ | Herbert Y Lin (Program in Membrane Biology/Nephrology, Department of Medicine, Massachusetts General Hospital and Harvard Medical School, Boston, Massachusetts); *Li et al., 2018* | | |
| Other | BZX800 fluorescent microscope | Keyence | RRID:SCR_023617 | Microscope used for I.F. |

*Appendix 1 Continued on next page*

*Appendix 1 Continued*

| Reagent type (species) or resource | Designation | Source or reference | Identifiers | Additional information |
|---|---|---|---|---|
| Other | GentleMACS Dissociator | Miltenyi Biotec | RRID:SCR_020272 | See 'Flow cytometry' in 'Materials and methods' section |
| Other | PE Annexin V | BioLegend | Cat# 640907 | Fluorescent stain for cells undergoing apoptosis |
| Other | Soluble tetrameric B8R$_{20-27}$/H2-K$^b$ complex | NIH Tetramer Core Facility | | For identifying CD8+T cells that respond to B8R$_{20-27}$ peptide |
| Other | TotalSeq-C Anti-Mouse Hashtags 1–10 | BioLegend | Cat #155861–155879 | Sample identifiers to allow pooling in scRNAseq |
| Other | FcBlock | Bio X Cell | Cat #BE0307 | To reduce false positives in flow cytometry |
| Other | NovaSeq S2 platform | Illumina | RRID:SCR_024569 | Used for scRNA sequencing |
| Other | ImmGenT TotalSeq-C custom mouse panel | BioLegend | Part no. 900004815 | Custom hashtags for scRNA-seq sample identification |
| Peptide, recombinant protein | OVA, N4, SIINFEKL | Genscript/Fisher | gene synthesis RP10161 | |
| Peptide, recombinant protein | Y3, SIYNFEKL | Genscript/Fisher gene synthesis RP10161 | gene synthesis RP10161 | |
| Software, algorithm | BZ-H4A Advanced Analysis Software | Keyence | RRID:SCR_017375 | |
| Software, algorithm | Adobe Photoshop (v6) | Adobe | RRID:SCR_014199 | |
| Software, algorithm | 10 x Genomics 5' v2 platform with Feature Barcoding for Cell Surface Protein and Immune Receptor Mapping | 10x Genomics | RRID:SCR_019326 | |
| Software, algorithm | ImmgenT single-cell RNA sequencing and processing | ImmgenT | *Casey and Zemmour, 2025* | https://www.immgen.org/ImmGenT/ |
| Software, algorithm | CellRanger | 10x Genomics | RRID:SCR_023221 | |
| Software, algorithm | Seurat v4.1 | Satija lab and collaborators | RRID:SCR_007322 | |
| Software, algorithm | Prism v10 | GraphPad | RRID:SCR_002798 | |
| Software, algorithm | FlowJo v10 | FlowJo | RRID:SCR_008520 | |
| Strain, strain background (Vaccinia virus) | VV | Dr. J. Yewdell, National Institute of Allergy and Infectious Diseases | Vaccinia virus-Western Reserve strain | |
| Strain, strain background (Vaccinia virus) | VV-OVA | Dr. J. Yewdell, National Institute of Allergy and Infectious Diseases | Vaccinia virus-Western Reserve strain expressing OVA$_{257-264}$ | |

