## [Editor Report · eLife Assessment]

This manuscript advances the prior finding that antigen recognition in the skill helps establish skin resident memory in CD8 T cells by elucidating the role of TGFBRIII in regulating CD8+ TRM skin persistence upon topical antigen exposure. Key novelty of the your work lies in generation and use of the CD8+ T cell-specific TGFBRIII knockout model, which allows them to demonstrate the role of TGFBRIII in fine tuning the degree of CD8+ T cell skin persistence and that TGFBRIII expression is promoted by CD8+ TRM encountering their cognate antigen upon initial skin entry. This is an **important** finding and is supported by **convincing** evidence. There are concerns about the use of FTY720 and the need to establish active TGFbeta limiting conditions to further test this working model.

---

## [Referee Report · Reviewer #1 (Public review)]

Summary:

Weiss et. al. seek to delineate the mechanisms by which antigen-specific CD8+ T cells outcompete bystanders in the epidermis when active TGF-b is limiting, resulting in selective retention of these cells and more complete differentiation into the TRM phenotype.

Strengths:

They begin by demonstrating that at tissue sites where cognate antigen was expressed, CD8+ T cells adopt a more mature TRM transcriptome than cells at tissue sites where cognate antigen was never expressed. By integrating their scRNA-Seq data on TRM with the much more comprehensive ImmGenT atlas, the authors provide a very useful resource for future studies in the field. Furthermore, they conclusively show that these "local antigen-experienced" TRM have increased proliferative capacity and that TCR avidity during TRM formation positively correlates with their future fitness. Finally, using an elegant experimental strategy, they establish that TCR signaling in CD8+ T cells in the epidermis induces TGFBRIII expression, which likely contributes to endowing them with a competitive advantage over antigen-inexperienced TRM.

Weaknesses:

The main weakness in this paper lies in the authors' reliance on a single experimental model to derive conclusions on the role of local-antigen during the acute phase of the response by comparing T cells in model antigen-vaccinia virus (VV-OVA) exposed skin to T cells in contralateral skin exposed to DNFB 5 days after the VV-OVA exposure. In this setting, antigen-independent factors may contribute to the difference in CD8+ T cell number and phenotype at the two sites. For example, it was recently shown that very early memory precursors (formed 2 days after exposure) are more efficient at seeding the epithelial TRM compartment than those recruited to skin at later times (Silva et al, Sci Immunol, 2023). DNFB-treated skin may therefore recruit precursors with reduced TRM potential. In addition, TRM-skewed circulating memory precursors have been identified (Kok et al, JEM, 2020), and perhaps VV-OVA exposed skin more readily recruits this subset compared to DNFB-exposed skin. Therefore, when the DNFB challenge is performed 5 days after vaccinia virus, the DNFB site may already be at a disadvantage in the recruitment of CD8+ T cells that can efficiently form TRM. In addition, CD8+ T cell-extrinsic mechanisms may be at play, such as differences in myeloid cell recruitment and differentiation or local cytokine and chemokine levels in VV-infected and DNFB-treated skin that could account for differences seen in TRM phenotype and function between these two sites. Although the authors do show that providing exogenous peptide antigen at the DNFB-site rescues their phenotype in relation to the VV-OVA site, the potential antigen-independent factors distinguishing these two sites remain unaddressed. In addition, there is a possibility that peptide treatment of DNFB-treated skin initiates a second phase of priming of new circulatory effectors in the local-draining lymph nodes that are then recruited to form TRM at the DFNB-site, and that the effect does not solely rely on TRM precursors at the DNFB-treated skin site at the time of peptide treatment. These concerns are somewhat alleviated by the fact that in a prior publication (PMID: 33212014), the group has already established a role for local antigen encounter in skin in a setting where they compared contralateral ears infected with VV-OVA and VV expressing an irrelevant antigen.

Secondly, although the authors conclusively demonstrate that TGFBRIII is induced by TCR signals and required for conferring increased fitness to local-antigen experienced CD8+ TRM compared to local antigen-inexperienced cells, this is done in only one experiment, albeit repeated 3 times. The data suggest that antigen encounter during TRM formation induces sustained TGFBRIII expression that persists during the antigen-independent memory phase. It remains however, unclear why only antigen encounter in skin, but not already in the draining lymph nodes, induces sustained TGFBRIII expression. Further characterizing the dynamics of TGFBRIII expression on CD8+ T cells during priming in draining lymph nodes and over the course of TRM formation and persistence may shed more light on this question. Probing the role of this mechanism at other sites of TRM formation would also further strengthen their conclusions and enhance the significance of this finding.

A minor caveat of the study pertains to the use of FTY720 to block T cell egress from lymphoid tissues and thereby prevent a contribution of circulating memory OT-I T cells to the local recall response in skin. Since the half-life of FTY720 is less than a day in mice, its effects wear off rapidly. In their experiments, the authors discontinued treatment at the time of re-challenge, which may have allowed circulating T cells to contribute to the local recall response in skin, limiting the interpretability of the results somewhat. This concern is alleviated by the use of a second method (anti-Thy1.1-depleting antibodies) to eliminate circulating memory cells. For the benefit of readers intending to use this experimental strategy, it should however, be noted that FTY720 needs to be dosed continually (e.g. 3x/week at an appropriate dose) in order to sustain its effect.

---

## [Referee Report · Reviewer #2 (Public review)]

Summary:

The authors set out to dissect the mechanistic basis of their previously published finding that encountering cutaneous antigen augments the persistence of CD8+ memory T cells that enter skin (TRM) (Hirai et al., 2021, Immunity). Here they use the same murine model to study the fate of CD8+ T cells after antigen-priming in the lymph nodes, (1) those that re-encounter antigen in the skin via vaccinia virus (VV) versus (2) those that do not encounter antigen in skin but rather are recruited via topical dinitrofluorobenzene (DNFB) (so-called "bystander TRM"). The authors' previous publication establishes that this first group of CD8+ TRM has a persistence advantage over bystander TRM under TGFb-limiting conditions. The current paper advances this finding by elucidating the role of TGFBR3 in regulating CD8+ TRM skin persistence upon topical antigen exposure. Key novelty of the work lies in generation and use of the CD8+ T cell-specific TGFBR3 knockout model, which allows them to demonstrate the role of TGFBR3 in fine tuning the degree of CD8+ T cell skin persistence and that TGFBR3 expression is promoted by CD8+ TRM encountering their cognate antigen upon initial skin entry. Future work directly measuring active TGFb in the skin under different conditions would help identify physiologic scenarios which yield active TGFb-limiting conditions, thus establishing physiologic relevance.

Strengths:

Technical strengths of the paper include (1) complementary imaging and flow cytometry analyses, (2) integration of their scRNA-seq data with the existing CD8+ TRM literature via pathway analysis, and (3) use of orthogonal models where possible. Using a vaccina virus (VV) model, with and without ovalbumin (OVA), the authors investigate how topical antigen exposure and TCR strength regulate CD8+ TRM skin recruitment and retention. The authors use both FTY720 and a Thy1.1 depleting antibody to demonstrate that skin CD8+ TRM expand locally following both a primary and secondary recall response to topical OVA application.

A conceptual strength of the paper is the authors' observation that TCR signal strength upon initial TRM tissue entry helps regulate the extent of their local re-expansion on subsequent antigen re-exposure. They achieved this by applying peptides of varying affinity for the OT-I TCR on the DNFB-exposed flank in tandem with initial VV-OVA + DNFB treatment. They then measured TRM expansion after OVA peptide rechallenge, revealing that encountering a higher affinity peptide upon skin entry leads to greater subsequent re-expansion. Additionally, by generating an OT-I Thy1.1+ E8i-creERT2 huNGFR Tgfbr3fl/fl (Tgfbr3∆CD8) mouse, the authors were able to elucidate a unique role for TGFBR3 in CD8+TRM persistence when active TGFb in skin is limited.

Weaknesses:

Overall, the authors' conclusions are well supported although there are some instances where additional controls, experiments, or clarifications would add rigor. The conclusions regarding skin localized TCR signaling leading to increased skin CD8+ TRM proliferation in-situ and increased TGFBR3 expression would be strengthened by assessing skin CD8+ TRM proliferation and TGFBR3 expression in models of high versus low avidity topical OVA-peptide exposure. The authors could further increase the impact of the paper by fully exploring whether TGFBR3 is regulated at the RNA or protein level; analysis of scRNAseq data included in the rebuttal did show an increase in Tgfbr3 RNA transcript levels in VV-treated compared to DNFB-treated back skin.

Quantification of the skin TRM population relies primarily on imaging analysis, which the authors indicate is more sensitive and consistent for quantifying this population. While flow cytometry is used to perform some phenotyping of TRMs, there remain some missed opportunities for more extensive analysis of markers expressed by this population. Finally, quantifying right and left skin draining lymph node CD8+ T cell numbers would clarify the skin specificity and cell trafficking dynamics of the authors' model.

This work heavily utilizes models developed and defined in previously published work (Hirai, T., et al., Competition for Active TGFβ Cytokine Allows for Selective Retention of Antigen-Specific Tissue- Resident Memory T Cells in the Epidermal Niche. Immunity, 2021. 54(1): p. 84-98.e5). Rather than repeating control experiments for this manuscript, the authors reference data included in this prior work. Thus, readers interested in a more in-depth understanding of these tools and concepts would be encouraged to read both papers.

---

## [Author Response]

The following is the authors’ response to the original reviews.

**Reviewer #1 (Public review):**
Weaknesses:The main weakness in this paper lies in the authors' reliance on a single model to derive conclusions on the role of local antigen during the acute phase of the response by comparing T cells in model antigen-vaccinia virus (VV-OVA) exposed skin to T cells in contralateral skin exposed to DNFB 5 days after the VV-OVA exposure. In this setting, antigen-independent factors may contribute to the difference in CD8+ T cell number and phenotype at the two sites. For example, it was recently shown that very early memory precursors (formed 2 days after exposure) are more efficient at seeding the epithelial TRM compartment than those recruited to skin at later times (Silva et al, Sci Immunol, 2023). DNFB-treated skin may therefore recruit precursors with reduced TRM potential. In addition, TRM-skewed circulating memory precursors have been identified (Kok et al, JEM, 2020), and perhaps VV-OVA exposed skin more readily recruits this subset compared to DNFB-exposed skin. Therefore, when the DNFB challenge is performed 5 days after vaccinia virus, the DNFB site may already be at a disadvantage in the recruitment of CD8+ T cells that can efficiently form TRM. In addition, CD8+ T cell-extrinsic mechanisms may be at play, such as differences in myeloid cell recruitment and differentiation or local cytokine and chemokine levels in VV-infected and DNFB-treated skin that could account for differences seen in TRM phenotype and function between these two sites. Although the authors do show that providing exogenous peptide antigen at the DNFB-site rescues their phenotype in relation to the VV-OVA site, the potential antigen-independent factors distinguishing these two sites remain unaddressed. In addition, there is a possibility that peptide treatment of DNFB-treated initiates a second phase of priming of new circulatory effectors in the local-draining lymph nodes that are then recruited to form TRM at the DFNB-site, and that the effect does not solely rely on TRM precursors at the DNFB-treated skin site at the time of peptide treatment.

Thank you for pointing out these potential caveats to our work. We have considered the possibility that late application of peptide or cell-extrinsic difference could affect the interpretation of our results. We would like to highlight that in our prior publication on this topic [1], we found that OT-1 responses in mice infected with VV-OVA and VV-N (irrelevant antigen) yielded the same responses as in our VV-OVA/DNFB models. In addition, in both our prior publication and our current manuscript, application of peptide to DNFB painted sites results in T_RM_ with a similar phenotype to those in the VV-OVA site. Thus, we are confident that it is the presence of cognate antigen in the skin that drives the augmented T_RM_ fitness that we observe.

Secondly, although the authors conclusively demonstrate that TGFBRIII is induced by TCR signals and required for conferring increased fitness to local-antigen-experienced CD8+ TRM compared to local antigen-inexperienced cells, this is done in only one experiment, albeit repeated 3 times. The data suggest that antigen encounter during TRM formation induces sustained TGFBRIII expression that persists during the antigen-independent memory phase. It remains unclear why only the antigen encounter in skin, but not already in the draining lymph nodes, induces sustained TGFBRIII expression. Further characterizing the dynamics of TGFBRIII expression on CD8+ T cells during priming in draining lymph nodes and over the course of TRM formation and persistence may shed more light on this question. Probing the role of this mechanism at other sites of TRM formation would also further strengthen their conclusions and enhance the significance of this finding.

This is an intriguing point. We do not understand why expression of TGFbR3 in T_RM_ required antigen encounter in the skin if T_RM_ at all sites clearly have encountered antigen during priming in the LN. We speculate that durable TGFbR3 expression may require antigen encounter in the context of additional cues present in the periphery or only once cells have committed to the T_RM_ lineage. A more detailed characterization of the dynamics of TGFbR3 expression in multiple tissues would be informative and represents a promising future direction for this project. We note that to robustly perform these experiments a reporter mouse would likely be a requirement.

**Reviewer #2 (Public review):**
Weaknesses:Overall, the authors' conclusions are well supported, although there are some instances where additional controls, experiments, or clarifications would add rigor. The conclusions regarding skin-localized TCR signaling leading to increased skin CD8+ TRM proliferation in-situ and increased TGFBR3 expression would be strengthened by assessing skin CD8+ TRM proliferation and TGFBR3 expression in models of high versus low avidity topical OVA-peptide exposure.

Thank you for these helpful suggestions. We did not attempt these experiment as we were concerned that given the relatively modest expansion differences observed with the APL that resolving differences in TGFbR3 and BrdU would prove unreliable. However, this is something that we could attempt as we continue working on this project.

The authors could further increase the novelty of the paper by exploring whether TGFBR3 is regulated at the RNA or protein level. To this end, they could perform analysis of their single-cell RNA sequencing data (Figure 1), comparing Tgfbr3 mRNA in DNFB versus VV-treated skin.

As discussed above, a more detailed analysis of TGFbR3 regulation is of great interest. These experiments would likely require the creation of additional tools (e.g. a reporter mouse) to provide robust data. However, as suggested, we have re-analyzed our scRNAseq looking for expression of Tgfbr3. Pseudobulk analysis of cells isolated from VV or DNFB sites suggests that Tgfbr3 appears to be elevated in antigen-experienced TRM at steady-state (Author response image 1).

**Author response image 1. sa3fig1:** Pseudobulk analysis by average gene expression of Tgfbr3 in cells isolated from either VV or DNFB treated flanks, divided by the average gene expression of Tgfbr3 in naïve CD8 T cells from the same dataset.

For clarity, when discussing antigen exposure throughout the paper, it would be helpful for the authors to be more precise that they are referring to the antigen in the skin rather than in the draining lymph node. A more explicit summary of some of the lab's previous work focused on CD8+ TRM and the role of TGFb would also help readers better contextualize this work within the existing literature on which it builds.

We appreciate this feedback, and we have clarified this in the text.

For rigor, it would be helpful where possible to pair flow cytometry quantification with the existing imaging data.

Thank you for these suggestions. In terms of quantification of number of T_RM_ by flow cytometry, we have previously demonstrated as much as a 36-fold decrease in cell count when compared to numbers directly visualized by immunofluorescence [1]. Thus, for enumeration of T_RM_ we rely primarily on direct IF visualization and use flow cytometry primarily for phenotyping.

Additional controls, namely enumerating TRM in the opposite, untreated flank skin of VV-only-treated mice and the treated flank skin of DNFB-only treated mice, would help contextualize the results seen in dually-treated mice in Figure 2.

Without a source of inflammation (e.g. VV infection of DNFB) we see very few T_RM_in untreated skin. A representative image is provided (Author response image 2). A single DNFB stimulation does not recruit any CD8+ T cells to the skin without a prior sensitization [2].

**Author response image 2. sa3fig2:** Representative images of epidermal whole mounts of VV treated flank skin, and an untreated site from the same mouse isolated on day 50 post infection and stained for CD8a.

In figure legends, we suggest clearly reporting unpaired T tests comparing relevant metrics within VV or DNFB-treated groups (for example, VV-OVA PBS vs VV-OVA FTY720 in Figure 3F).

Thank you for this suggestion. The figure legends have been amended.

Finally, quantifying right and left skin draining lymph node CD8+ T cell numbers would clarify the skin specificity and cell trafficking dynamics of the authors' model.

We quantified the numbers of CD8 T cells in left and right skin draining lymph nodes by flow cytometry in mice at day 50 post VV infection DNFB-pull. We observe similar numbers of cells at both sites (Author response Image 3).

**Author response image 3. sa3fig3:** Quantification of total number of CD8+ T cells in left and right inguinal lymph nodes. Each symbol represents paired data from the same individual animal, and this is representative of 3 separate experiments.

**Reviewer #1 (Recommendations for the authors):**
(1) Figures 1D and S1C demonstrate that 80-90 % of TRM at both VV and DNFB sites express CD103+. In contrast, the sequencing data suggests the TRM at the VV site has much higher Itgae expression. Also, clusters 3 and 4, which express significantly more Itgae than all other clusters, together comprise only ~30% of CD8+ T cells at the VV-infected skin site. How can these discrepancies between transcript and protein expression be explained?

Thank you for these excellent comments. T_RM_ at both VV and DNFB sites appear to express similarly high levels of CD103 protein in both the OT-I system as we previously published [1] and in a polyclonal system using tetramers. The lower penetrance of Itgae expression in the scRNAseq data we attribute to a lack of sensitivity which is common with this modality. However, the relative increased expression of Itgae in clusters 3 and 4 is interesting and may suggest increased Itgae production/stability. However, in the absence of any effect on protein expression, we chose not to focus on these mRNA differences.

(2) For the experiments in Figure 3D, in order to exclude a contribution from circulating memory cells, FTY720 should have been administered during the duration of, not prior to, the initiation of the recall response. The effect of FTY720 wears off quickly, so the current experimental setting likely allows for circulating cells to enter the skin. This concern is mitigated by the results of anti-Thy1.1 mAb treatment, but documenting the experiment as in Figure D will likely be confusing to readers.

Thank you for this comment. We relied on the literature indicating that the half-life of FTY720 in blood is longer than 6 days [3-5]. However, on reviewing this again, there are other reports suggesting a lower halflife. Thank you for pointing out this potential caveat. As mentioned above, we do not think this affects the interpretation of our data as similar results were obtained with anti-Thy1.1

(3) Similar to what is described in the weaknesses section, the data on TGFBRIII expression is lacking. When is TGFBRIII induced? In the LN during primary activation and it is then sustained by a secondary antigen exposure at the peripheral target tissue site? Or is it only induced in the peripheral tissue, and there is interesting biology to uncover in regard to how it is induced by the TCR only after secondary exposure, etc.?

Thank you for these comments. As discussed above, a more detailed analysis of TGFbR3 regulation is of great interest. These experiments would likely require the creation of additional tools (e.g. a reporter mouse) to provide robust data and are part of our future directions.

(4) As described in the weakness section, there could be TCR-independent differences between the VV-OVA and DNFB sites that lead to phenotypic changes in the TRMs that are formed there, both CD8+ T cell-intrinsic (kinetics; with regard to time after initial priming) and extrinsic (microenvironmental differences due to the nature of the challenge, recruited cell types, cytokines, chemokines, etc.). Since the authors report the use of both VV and VV-ova, we recommend an experimental strategy that controls for this by challenging one site with VV and another with VV-OVA concomitantly, followed by repeating the key experiments reported in this manuscript.

As discussed above, we have previously published a very similar experiment using VV-OVA and VV-N infection on opposite flanks [1].

(5) In Figure 6J please indicate means and provide more of the statistics comparing the groups (such as comparing VV-WT vehicle to VV-KO vehicle etc.), and potentially display on a linear scale as with all of the other figures looking at cells/mm2 to help convince the reader of the conclusions and support the secondary findings mentioned in the text such as "Notably, numbers of Tgfbr3ΔCD8 TRM in cohorts treated with vehicle remained at normal levels indicating that loss of TGFβRIII does not affect TRM epidermal residence in the steady state" despite it looking like there is a decrease when looking at the graph.

We appreciate the feedback on the readability of this figure, and so have updated figure 6J to be on a linear scale and added additional helpful statistics to the figure legend. The difference between Tgfbr3^WT^ and Tgfbr3^∆CD8^ at steady state is excellent point, and we agree that there could to be a trend towards reduction in the huNGFR+ T_RM_ across both groups, even without CWHM12 administration. However, we did not see statistically significant reductions in steady-state Tgfbr3^∆CD8^ T_RM_, but the slight reduction in both VV-OVA and DNFB treated flanks suggests that TGFßRIII may play a role in steady-state maintenance of all T_RM_. Perhaps with more sensitive tools to better visualize TGFßRIII expression, we could identify stepwise upregulation of TGFßRIII depending on TCR signal strength, possibly starting in the lymph node. We have also amended our description of this figure in the text, to allow for the possibility that a low, but under the level of detection amount of TGFßRIII could play a role in steady-state maintenance of both local antigen-experienced and bystander T_RM_.

Minor points:(1) In describing Figure 4B, the term "doublets" for pairs of connected dividing cells is confusing.

Thank you for this comment, the term has been revised to “dividing cells” in the text and figure.

(2) Figure legend 4F: BrdU is not "expressed" .

Very true, it has been changed to “incorporation”.

(3) Do CreERT2 and/or huNGFR expressed by transferred OT-I cells act as foreign antigens in C57BL/6 mice, potentially causing elimination of circulating memory cells? If that were the case, this would not necessarily confound the read-out of TRM persistence studied here, since skin TRM are likely protected from at least antibody-mediated deletion and their numbers are not maintained by recruitment of circulating cells at stead-state. However, it would be useful to be aware of this potential limitation of this and similar models.

Thank you for raising the important technical concern. In our prior work [1] and this work, we monitor the levels of transferred OT-I cells in the blood over time. We have not observed rejection of huNGFR+ cells. We also note that others using the same system have also not observed rejection [6].

(4) In Figure 6J, means or medians should be indicated

This has been updated in Figure 6J.

(5) Using the term "antigen-experienced" to specifically refer to TRM at the VV site could be confusing, since those at the DNFB site are also Ag-experienced (in the LN draining the VV skin site).

We agree that it is a challenging term, as all T_RM_ are memory cells. That is why in the text we refer to T_RM_ isolated from the VV site as “local antigen experienced T_RM_.”, to try to distinguish them from bystanders that did not experience local antigen.

(6) The Title essentially restates what was already reported in the authors' prior study. If the data supporting the TGFBRIII-mediated mechanism is studied in more depth, maybe adding this aspect to the title may be useful?

Thank you for this suggestion. I think the current title is probably most suitable for the current manuscript but we are willing to change it should the editors support an alternative title.

**Reviewer #2 (Recommendations for the authors):**
(1) Definition of bystander CD8+ TRM: The first paragraph of the introduction defines CD8+ TRM. To improve the clarity of this definition, we suggest being explicit that bystander TRM experience cognate antigen in the SDLNs but, in contrast to other TRM, do not experience cognate antigen in the skin.

Thank you, we have clarified this is in the text.

(2) Consider softening the language when comparing the efficiency of CD8+ recruitment of the skin between DNFB and VV-treated flanks. For example, substitute "equal efficiency" with "comparable efficiency" since it is difficult to directly compare the extent of inflammation between viral and hapten-based treatments.

We have adjusted this terminology throughout the paper.

(3) Throughout figure legends, we appreciate the indication of the number of experimental repeats performed. We suggest, either through statistics or supplemental figures, demonstrating the degree of variability between experiments to aid readers in understanding the reproducibility of results.

Thank you for this suggestion. In key figures we show data from individual mice across multiple experiments. Thus, inter-experiment variability is captured in our figures.

(4) Figure 1:a) Add control mice treated with either vaccinia virus or DNFB and harvest back skin at day 52 to demonstrate baseline levels of polyclonal and B8R tetramer-positive CD8s in the epidermis. These controls would clarify the background CD8+ expansion that might occur in DNFB-treated mice in the absence of vaccinia virus.

This point was addressed above.

b) Figure 1: It would be helpful to see the %Tet+ population specifically in the CD103+ population, recognizing that the majority of the CD8+ from the skin are CD103+.

We did look only at CD103+ CD8 T cells from the skin for our tetramer analysis, so this has been clarified in the figure legend.

c) Provide a UMAP, very similar to 1H, where CD8+ T cells, vaccinia virus, and DNFB-treated flanks are overlaid.

Thank you for this suggestion. A UMAP combining aspects of 1G (cell types from the whole ImmgenT dataset) with 1H (our data) results in a figure that is very difficult to interpret. Thus, we have separated cell types across the entire ImmgenT data set (e.g. CD8+ T cells) and our data into 2 separate panels.

d) 1D: left flow plot has numbered axis while the right flow plot does not.

Thank you, this has been fixed.

(5) Figure 2:a) In the figure legend, define what is meant by the grey line present in Figures 2C and 2D.

This has been updated in the figure legend.

b) Edit the Y axis of 2C and 2D to specify the TRM signature score.

This has been updated in the figure.

c) Include panel 1D from 1S into Figure 2 to help clarify for the reader what genes are expressed in the 0 - 5 clusters.

We appreciate the feedback, but we found the heatmap made the figure look too busy, so we feel comfortable keeping it available within supplemental figure 1.

d) In body of text explicitly discuss that the TRM module used to calculate a signature score was created using virus infection modules (HSV, LCMV and influenza) and thus some of the transcriptional similarity between the authors vaccinia virus treated CD8+ TRM and the TRM module might be due to viral infection rather than TRM status.

Thank you for this comment. We have now emphasized this point in the text.

(6) Figure 3:a) If there are leftover tissue sections, it would be optimal to show specific staining for CD103. We recognize that this data has been previously published by the lab, but it would be ideal to show it once in this paper.

Unfortunately, we do not have leftover tissue sections, so we are unable to measure CD103 by I.F. in these experiments.

b) If you did collect skin draining lymph nodes in the Thy1.1 depletion model, it would be nice to see flow data showing the depletion effects in the skin draining lymph nodes in addition to the blood.

Unfortunately, we did not collect the skin draining lymph nodes, and do not have that data for the relevant experiments.

c) Figure 3 F & G: Perform a T-test comparing vaccinia virus PBS to FTY720 and isotype to anti-Thy1.1 within the same treatment group. Showing no significance with these two comparisons would strengthen the authors' claims. Statistics can be described in legend.

We have included this analysis in the figure legend.

(7) Figure 4:a) It would be helpful to have the CD69+/CD103+ population in this model discussed/defined more. The CD69 expression seen in 4E is lower than the reviewers would've predicted, and it would be interesting to see CD103 expression as well.

We have found that generally CD103 is a stronger marker for in the skin by flow, as CD69 staining is somewhat less robust in the colors we have chosen. By way of example, we present gating we did upstream in that experiment, gated previously on liveCD45+CD3+CD8+ events (Author response image 4)

**Author response image 4. sa3fig4:** Representative flow cytometric plots showing CD69 and CD103 expression in gated live CD45+CD8+CD90.1+ cells isolates from VV-OVA or DNFB treated flanks.

(8) Figure 5:a) Define APL and its purpose in both the body of text and the figure legend.

We have clarified this in the text and the figure legend.

b) Using in-vivo BrdU, compare proliferation between high avidity N4 and low avidity Y3 OVA-peptide at the primary recall timepoint.

We considered this, but due to the lack of sensitivity of the BrdU incorporation and the relatively subtle phenotype of the Y3, we did not think the assay would be sensitive enough to identify differences.

(9) Figure 6:a) Compare TGFBR3 expression in CD8+ T cells from mice receiving high avidity N4 versus low avidity Y3 OVA-peptide at the primary recall timepoint.

This point was discussed above.

b) Either 1) examine TGFBR3 mRNA expression in VV vs DNFB skin from scRNA-seq dataset or 2) perform a qPCR on epidermal CD8+ T cells from mice receiving high avidity N4 versus low avidity Y3 at the primary recall timepoint. This would help distinguish whether TGFBR3 regulation occurs at the mRNA versus protein level.

This point has been discussed above.

c) Figure 6A: Not required, but it seems like the TGFBR3 gate could be shifted to the right a bit.

The gates were set using FMO.

d) Figure 6C: What comparison is the asterisk indicating significance referring to?

It is the Dunnett’s test comparing VV-OVA to DNFB and untreated skin, the figure has been amended to clarify this point.

e) Figure 6: To increase the rigor of the claim that CWHM12 is creating a TGFb limiting condition, the authors could either 1) perform an ELISA or cell-based assay measuring active TGFb, 2) recapitulate results of 6J using monoclonal antibody against avb6 as done in Hirai et al., 2021, Immunity., or 3) examine Tgfbr3 mRNA expression in your single cell RNAseq data, comparing cluster 0 and cluster 3.

We are pleased to have the opportunity to show Tgfbr3 mRNA, which is above in figure R1.

(10) Material and methods:Specify how the localization of the back skin used for imaging was made consistent between the right and left flanks.

We have updated this methodology in the text.

Literature Cited

(1) Hirai, T., et al., Competition for Active TGFβ Cytokine Allows for Selective Retention of Antigen-Specific Tissue- Resident Memory T Cells in the Epidermal Niche. Immunity, 2021. 54(1): p. 84-98.e5.

(2) Manresa, M.C., Animal Models of Contact Dermatitis: 2,4-Dinitrofluorobenzene-Induced Contact Hypersensitivity, in Animal Models of Allergic Disease: Methods and Protocols, K. Nagamoto-Combs, Editor. 2021, Springer US: New York, NY. p. 87-100.

(3) Müller, H.C., et al., The Sphingosine-1 Phosphate receptor agonist FTY720 dose dependently affected endothelial integrity in vitro and aggravated ventilator-induced lung injury in mice. Pulmonary Pharmacology & Therapeutics, 2011. 24(4): p. 377-385.

(4) Nofer, J.-R., et al., FTY720, a Synthetic Sphingosine 1 Phosphate Analogue, Inhibits Development of Atherosclerosis in Low-Density Lipoprotein Receptor–Deficient Mice. Circulation, 2007. 115(4): p. 501-508.

(5) Brinkmann, V., et al., Fingolimod (FTY720): discovery and development of an oral drug to treat multiple sclerosis. Nat Rev Drug Discov, 2010. 9(11): p. 883-97.

(6) Andrews, L.P., et al., A Cre-driven allele-conditioning line to interrogate CD4^+^ conventional T cells. Immunity, 2021. 54(10): p. 2209-2217.e6.